# Adaptive Layer Sparsity for Large Language Models via Activation Correlation Assessment

**Wei Li[1], Lujun Li[2†], Mark Lee[1*], Shengjie Sun[3]**
[1]University of Birmingham
[2]Hong Kong University of Science and Technology, [3]AISpeech Co., Ltd.
WXL885@student.bham.ac.uk, lilujunai@gmail.com, M.G.Lee@bham.ac.uk
shengjie.sun@aispeech.com
*

## Abstract

Large Language Models (LLMs) have revolutionized the field of natural language processing with their impressive capabilities. However, their enormous size presents challenges for deploying them in real-world applications. Traditional compression techniques, like pruning, often lead to suboptimal performance due to their uniform pruning ratios and lack of consideration for the varying importance of features across different layers. To address these limitations, we present a novel Adaptive Layer Sparsity (ALS) approach to optimize LLMs. Our approach consists of two key steps. Firstly, we estimate the correlation matrix between intermediate layers by leveraging the concept of information orthogonality. This novel perspective allows for a precise measurement of the importance of each layer across the model. Secondly, we employ a linear optimization algorithm to develop an adaptive sparse allocation strategy based on evaluating the correlation matrix. This strategy enables us to selectively prune features in intermediate layers, achieving fine-grained optimization of the LLM model. Considering the varying importance across different layers, we can significantly reduce the model size without sacrificing performance. We conduct extensive experiments on publicly available language processing datasets, including the LLaMA-V1|V2|V3 family and OPT, covering various benchmarks. Our experimental results validate the effectiveness of our ALS method, showcasing its superiority over previous approaches. The performance gains demonstrate its potential for enhancing LLMs' efficiency and resource utilization. Notably, our approach surpasses the state-of-the-art models Wanda and SparseGPT, showcasing its ability to excel even under high sparsity levels. Codes at: https://github.com/lliai/ALS.

## 1 Introduction

Large language models (LLMs) [62, 49, 3] have demonstrated remarkable performance in various natural language processing (NLP) [55, 54, 4] tasks. However, their size and computational requirements pose significant challenges for widespread adoption and deployment. To address these practical constraints, model compression techniques, such as weight pruning and quantization, can potentially reduce the size and computational requirements of LLMs.

The emergence of LLMs has revolutionized the field of NLP. However, despite their revolutionary impact, the massive scale and complexity of LLMs presents significant challenges for model compression. Conventional pruning methods [27, 19, 38, 19, 15, 59], which often require one or more

---

*Corresponding authors, † project lead with equal contribution.

38th Conference on Neural Information Processing Systems (NeurIPS 2024).

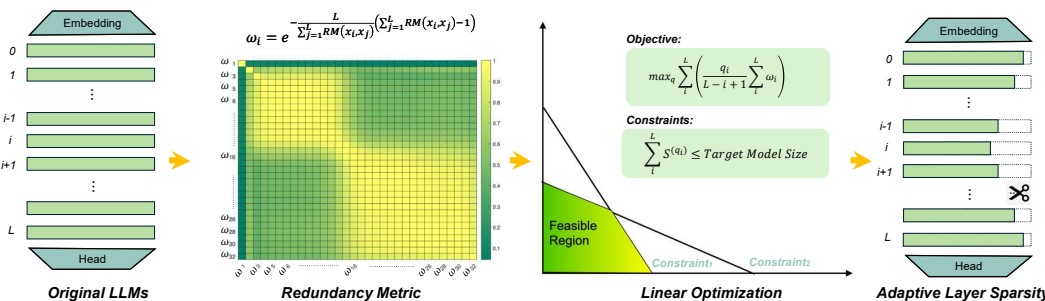

Figure 1: Overview of our framework. We first compute the sum of Redundancy Metric between layer $i$-th and other layers to construct objective function. Then, we solve a linear programming problem to optimize total sparsity ratios $S^{(q_i)}$ ($q_i$ is pre-layer sparsity) under constraints.

iterations of fine-tuning or retraining to preserve performance, have become impractical for LLMs due to the substantial computational cost and time required.

Due to the failure to the Magnitude approach to pruning [28] and other previous methods on LLMs, recent efforts such as SparseGPT [17], Wanda [46], DSOT [65], Pruning Large Language Models with BESA [57], and OWL [58] aim to address this challenge by reconstructing the layerwise outputs of LLMs. Specifically, SparseGPT introduces a technique for pruning less significant weights and reconstructing layerwise outputs based on an importance metric derived from the Hessian matrix. To reduce the computational overhead of SparseGPT, Wanda proposes a simplified strategy that relies solely on the product of weight and activation magnitudes for pruning. DSOT computes the reconstruction error incrementally for each layer, optimizing the intra-layer sparse configuration through further weight pruning or growth, which forms the basis for subsequent weight recovery and additional pruning operations. These methods adopt a training-free approach. In contrast, BESA [57] proposes learning the optimal pruning ratio within each layer through training, finding that considering the overall sparsity configuration within a layer enhances the performance of sparse models. However, this method primarily focuses on intra-layer sparsity configuration. It requires substantial training time, typically taking at least 5 hours on an A100-80G GPU, which is considerably slower than other training-free techniques [17, 46, 65]. Another notable method is OWL [58], which proposes a non-uniform layerwise sparsity technique that assigns different sparsity ratios based on the outlier ratio within each layer, leveraging the unique characteristic of LLMs where some features exhibit significantly larger magnitudes by tuning hyperparameters such as the outlier threshold and sparsity upper/lower bounds to obtain optimal parameter setting. Nevertheless, unlike the aforementioned methods, OWL relies heavily on empirical analysis without providing a solid theoretical foundation for its effectiveness.

However, existing methods have several significant drawbacks. First, for BESA, DSOT, and some traditional techniques, minimizing the layer-by-layer pruning error does not effectively mitigate the impact of pruning on model performance, as the pruning error accumulates across layers due to its inherent greedy nature [24] and may also become trapped in local optima [13, 22]. Second, LLM pruning methods such as Wanda, SparseGPT, and Magnitude apply uniform sparsity ratio to each layer, despite the significant variations in each layer's contribution to the final model performance [57, 65]. To achieve better performance for different layers, the sparsity needs to be manually adjusted for all layers. Third, for the newly proposed OWL method, more theoretical analysis is needed on why its outlier-based non-uniform sparsity outperforms uniform sparsity. Moreover, the choice of hyperparameters in OWL, such as the outlier threshold and sparsity upper/lower bounds, is sensitive to model performance, but their optimal ranges are not theoretically explained, and the effective ranges and thresholds are derived through manual tuning. Furthermore, the transferability of these hyperparameters across different datasets has yet to be systematically studied. Therefore, when applying OWL to new models, complex adjustments by hand must be performed to determine the potentially optimal parameter combination.

To address the multiple challenges of getting trapped in local optima, manually setting sparsity for all layers, and relying on empirical manual experiments to derive optimal sparsity ratios, we propose a simple, effective, and efficient method called Adaptive Layer Sparsity (ALS) for allocating sparsity ratios. The overall pipeline of our proposed method is illustrated in Fig. 1. This technique optimizes

the pruning rate across different layers. To the best of our knowledge, this is the first attempt to reformulate the sparsity allocation problem in LLMs as a linear programming problem. We tackle these challenges by constructing an objective function and constraints. The constraint of the linear programming problem is that the total number of parameters should be less than the target model size. We then compute the independence matrix [25] at both the layer level and intra-layer component level based on the output or input features. According to our experiments, the independence between layers is positively correlated with model performance, as shown in Fig. 2 (c). Therefore, maximizing the independence between each layer of the model is considered our objective function. Unlike existing black-box optimization methods, we formulate this problem as a linear one that can be solved by any linear problem solver. This approach enables efficient global sparsity ratio allocation for LLMs ranging from 7B to 70B parameters on a single A100-80GB GPU. If the model scale is too large, reaching 160B, we can also perform multi-threaded computation on a CPU. For a 70B model, the global sparsity configuration can be obtained in just 20 minutes on an A100-80G GPU.

To rigorously assess the efficacy of ALS, we conducted extensive experiments on diverse LLMs, including LLaMA-V1 [49], LLaMA-V2 [50], LLaMA-V3 [1], and OPT [62] model families, with parameter counts ranging from 6.7 billion to 70 billion. In the main experiments, we evaluated the WikiText-2 perplexity and average accuracy on 7 zero-shot datasets at various sparsity ratios (20% to 70%) for LLaMA-V2 7B/13B (Table 1) and at 50% sparsity for all model families (Tables 2 and 3). Detailed results for each zero-shot dataset on LLaMA-V2 family models at 50% sparsity are presented in Table 4. The analysis experiments consist of 6 sets, examining the impact of calibration data, sparsity bounds setting, and model redundancy on performance (Fig. 2), as well as the influence of feature selection, standardization, and comparisons with Wanda and LoRA fine-tuning. Additional experiments including detailed of main experiments, analyses of granularity, decreasing functions, visualizations of layer redundancy, sparsity ratio allocation and comparison with OWL method are provided in the Appendix C and D. These experimental results unequivocally demonstrate that ALS consistently yields substantial performance improvements for sparse LLMs across various LLMs and downstream tasks.

## 2 Related Work

**Model Compression** method try to design efficient models and reduce the memory and computational requirements of LLMs. These methods can be categorized into quantization [42, 12, 35, 33], sparsification [17, 46, 10, 9] and distillation [56, 29, 30, 31, 32, 11, 53]. Quantization converts high bit-width weights and activations into compact, low bit-width representations, while sparsification increases the proportion of zero-valued elements in model weights. Our method optimizes LLM sparsification by strategically allocating sparsity across the model's layers to maximize performance and minimize computational overhead. In contrast to optimization-based compression techniques (*e.g.*, OMPQ [35]) for CNN models in vision tasks, our approach focuses on different LLM models and NLP tasks and devises various functions and strategies accordingly.

**Sparsity in LLMs** has garnered significant attention as a means to accelerate inference speed and reduce memory consumption by applying sparsity in the model weights or activations. sparsity techniques can be categorized into two main approaches: structured pruning [34, 23] and unstructured pruning [16, 64, 46, 63]. While the primary focus of these works lies in determining the pruning criteria, such as weight importance and pruning ratio, the enormous parameter scale of LLMs presents an additional challenge in terms of pruning efficiency. Conventional pruning methods [15, 59, 63, 23, 27, 19, 38, 19], dating back to the early work of Hassibi [20] in the 1990s, which successfully reduced model size and improved efficiency in deep learning architectures by removing redundant weights to create sparse and lightweight models, heavily rely on extensive retraining and are often infeasible for LLMs due to prohibitively high computational overhead and prolonged training times. To address this issue, researchers have developed LLM-specific pruning techniques that prioritize train-free and time efficiency. In the context of structured pruning, LLMpruner [34] explores the application of structured pruning to LLMs and employs LoRA to recover the performance of the pruned model. For unstructured pruning, SparseGPT [17] stands out as a notable method that draws inspiration from the Optimal Brain Surgeon (OBS) [20] approach, taking into account the impact of removing individual weights on the network reconstruction loss. SparseGPT introduces an efficient technique for estimating the Hessian matrix, enabling the application of the traditional OBS method to large-scale models. Another prominent unstructured pruning method, Wanda [46], employs a simple yet

effective strategy based on the product of weight and activation values to identify and eliminate less important weights, further enhancing the pruning speed. Despite these advancements, most existing methods adopt a uniform pruning rate across all layers, which may lead to suboptimal performance. In contrast, our approach introduces a novel layer adaptive pruning strategy that dynamically allocates sparsity based on the importance of each layer, effectively minimizing performance degradation while achieving high compression ratios.

**Sparsity Allocation in Network Pruning.** Conventional methods for achieving adaptive layer-wise sparsity in neural networks [14, 5, 26] often rely on a layer-by-layer pruning approach, where the objective is to minimize the sum of errors introduced in each layer. However, this greedy strategy [24] leads to the accumulation of errors across layers, resulting in suboptimal performance when directly adapted to LLMs. The extensive retraining required on vast datasets further amplifies the challenges of applying these techniques to LLMs. Recent efforts, such as BESA [57] and DSOT [65], have shifted focus to intra-block sparsity allocation, employing various strategies to optimize the sparsity distribution within individual blocks. Despite operating at a finer granularity, these methods fundamentally adhere to a layer-wise pruning paradigm, neglecting the importance of global sparsity allocation. Consequently, the resulting allocation may be locally optimal [13, 22] within each layer but globally suboptimal, potentially leading to solutions stuck in local optima. Recently, a new approach called OWL [58] attempts to address this issue by introducing a non-uniform layer-wise sparsity technique. This technique primarily relies on manually tuning the outlier threshold and sparsity upper/lower bounds (which are very small values and sensitive to performance) through extensive experimentation to obtain potentially optimal parameter configurations. Although OWL demonstrates the potential for improved sparsity allocation, it heavily depends on empirical analysis and fails to provide a solid theoretical foundation for its effectiveness, limiting its generalizability and robustness across different LLMs architectures and datasets.

## 3 Methodology

### 3.1 Preliminary

Pruning LLMs is a method that aims to obtain a sparse representation of the model by eliminating a predetermined fraction of the pre-trained weights. The primary objective is to minimize the divergence between the outputs generated by the sparse and dense models [21]. However, directly tackling this problem can be challenging due to the massive scale of LLMs. We discover that the mutual information entropy in Eq. 1 can effectively quantify the degree of discrepancy between different layers of the model.

$$I\left(x_i; x_j\right) = H\left(x_i\right) + H\left(x_j\right) - H\left(x_i, x_j\right) \tag{1}$$

Neural networks can be decomposed into a sequence of layers. In the decomposed form, we represent the neural network as $F = \{f_1, f_2, \ldots, f_L\}$, For a given random sample $x_0 \in \mathbb{R}^{d_0}$, let $x_i = f_i\left(f_{i-1}\left(\ldots f_1\left(x_0\right)\right)\right) \in \mathbb{R}^{d_i}$ represents the output of the random sample at the $i$-th layer.

Based on the previous definitions of the marginal entropies and joint entropy, the mutual information between $x_i$ and $x_j$ can be derived from Eq. 1 and formally defined as Eq. 2 [8].

$$I\left(x_i; x_j\right) = \int p\left(x_i, x_j\right) \log \frac{p\left(x_i, x_j\right)}{p\left(x_i\right) p\left(x_j\right)} dx_i dx_j \tag{2}$$

High mutual information between layers indicates redundancy, while low mutual information suggests that these layers have learned complementary representations [43]. When two variables $x_i$ and $x_j$ are independent, their mutual information is zero, i.e., $I(x_i; x_j) = 0$. According to information theory [41] and the Information Bottleneck (IB) theory [48], minimizing the mutual information between layers can reduce redundancy, remove irrelevant information, and enhance the overall representational capacity of the network. This "compression" of the representation enables the network to extract higher-level and more compact features, thereby reducing the reconstruction error. In summary, by sparsifying layers with higher mutual information and minimizing the mutual information throughout the entire network, the reconstruction error can be minimized.

## 3.2 Redundancy Metric

To approximate the mutual information between layers, we propose employing Monte Carlo sampling, thereby circumventing the need for intractable integrals. Specifically, we randomly select $N$ samples $x_0^{(1)}, x_0^{(2)}, \ldots, x_0^{(N)}$ from the training dataset, which follow a probability density function (PDF) $P(x)$. For each sample $x_0^{(n)}$, the outputs at the $i$-th and $j$-th layers are denoted as $x_i^{(n)}$ and $x_j^{(n)}$, respectively.

We can estimate the integral using the sample average: $\hat{I}(x_i; x_j) \approx \frac{1}{N} \sum_{n=1}^{N} \log \frac{p\left(x_i^{(n)}, x_j^{(n)}\right)}{p\left(x_i^{(n)}\right) p\left(x_j^{(n)}\right)}$.

$\hat{I}(x_i; x_j)$ is the estimated mutual information, $p\left(x_i^{(n)}, x_j^{(n)}\right)$ is the joint PDF, and $p\left(x_i^{(n)}\right)$ and $p\left(x_j^{(n)}\right)$ are the marginal PDFs. We aim to approximate these probability densities using kernel density estimation.

**Computing marginal and joint probability densities**: Based on the kernel density estimation [44], we can utilize it to estimate the probability density functions. This allows us to approximate the probability density functions using the features of the samples. Kernel density estimation is a non-parametric method for estimating the probability density function of a random variable. For instance, given a set of samples $Y = \{y_1, y_2, \ldots, y_N\}$, The kernel density estimate is defined as: $\hat{p}(y) = \frac{1}{Nh} \sum_{i=1}^{N} K\left(\frac{y - y_i}{h}\right)$, where $K(\cdot)$ is a kernel function, commonly used kernels include Gaussian, Epanechnikov, etc., and $h$ is the bandwidth parameter.

We apply kernel density estimation to compute the marginal and joint probability density functions of the samples' outputs at the $i$-th and $j$-th layers. The choice of the bandwidth parameter $h$ can be determined through cross-validation or other methods. However, to simplify our derivation, we can consider that in high-dimensional spaces, the influence of the kernel function $K$ is relatively insignificant. We are mainly focused on the ratio of relative densities [40, 60]. Therefore, the bandwidth parameter $h$ can be cancelled out. We can calculate the marginal and joint probability density functions of the samples' outputs at the $i$-th and $j$-th layers using kernel density estimation, which can be found in the Appendix.

**Monte Carlo Approximation of Mutual Information.** Substituting the kernel density estimates into the Monte Carlo approximation formula for mutual information and simplifying the expression using the feature matrix inner product approximation for the kernel function, as mentioned by Tschannen [51], we obtain:

$$\hat{I}(x_i; x_j) \approx \frac{1}{N} \sum_{n=1}^{N} \log \frac{\left\| x_i^{(n)T} x_j^{(n)} \right\|_F}{\left\| x_i^{(n)T} x_i^{(n)} \right\|_F \left\| x_j^{(n)T} x_j^{(n)} \right\|_F} \tag{3}$$

where $\| \cdot \|_F$ denotes the Frobenius norm. In this approximation, we employ the feature matrix inner product to approximate the kernel function, $K\left(\left(x_i^{(n)}, x_j^{(n)}\right), \left(x_i^{(k)}, x_j^{(k)}\right)\right) \approx \left\| x_i^{(n)T} x_j^{(n)} \right\|_F$. Similarly, for the marginal kernel functions is $K\left(x_i^{(n)}, x_i^{(k)}\right) \approx \left\| x_i^{(n)T} x_i^{(n)} \right\|_F$.

**Decreasing function**: Since mutual information has no general upper bound, its upper limit depends on the entropy of either $x_i$ or $x_j$. To address this, we can use decreasing functions to transform the range of mutual information, ensuring a bounded and more interpretable metric. For instance, using $e^{\hat{I}(x_i; x_j)}$ or a Gaussian function, we can redefine the measure as follows. Considering that a batch of data is fed into the model simultaneously and each layer output concurrently, we can omit the $\sum_{n=1}^{N}$ and $\frac{1}{N}$. Instead, we can use $X_i$ and $X_j$ to represent the calculations for the entire batch input. Applying $\left\| X_i^T X_j \right\| e^{\hat{I}(X_i; X_j)}$ as a decreasing function to Eq. 3. Therefore, we can derive the Redundancy Metric (RM) formula, where $RM(\cdot) \in [0, 1]$ according to the Cauchy-Schwarz inequality:

$$RM(X_i, X_j) = \frac{\left\| X_i^T X_j \right\|^2}{\left\| X_i^T X_i \right\| \left\| X_j^T X_j \right\|} \tag{4}$$

The decreasing function transforms the range of the RM formula such that a value of 0 indicates complete independence between layers, while 1 represents complete redundancy. This formulation

can serve as the objective function for maximization. The complete derivation process for this section, including the details of Eq. 4, is presented in Appendix B.

### 3.3 Linear Optimization

Our Redundancy Metric reveals the redundancy among layers in a neural network, guiding sparsity ratio allocation. Experiments on LLaMA2-13B with various sparsity configurations show a negative correlation between model redundancy and WikiText-2 perplexity (PPL). Model redundancy is defined as the sum of each layer's RM for the remaining layers, as depicted in Fig. 2 (c). Consequently, redundancy minimization is adopted as the objective function, incorporating model size constraints to formulate a linear programming problem that yields the optimal sparsity configuration.

**Intra-layer Sparsity Allocation.** For a given neural network, we construct a redundancy matrix $\mathbf{\Psi}$, where $\psi_{ij} = RM(x_i, x_j)$. The sum of non-diagonal elements for each row of the matrix is computed as $\rho_i = \sum_{j=1}^{L} \psi_{ij} - 1$. A smaller $\rho_i$ indicates stronger independence between $x_i$ and the outputs of other layers. We model this relationship using the monotonically decreasing function: $\omega_i = e^{-\frac{1}{\mu}\rho_i}$, where $\mu$ is a dynamic hyperparameter controlling the difference in sparsity ratios across layers, defined as $\frac{1}{n} \sum_{j=1}^{n} \psi_{ij}$, which smooths the descent speed (Fig. 8). The importance factor for the first $i$ layers is represented by $\omega_i$. With these components, we formulate the linear programming problem as follows:

$$\text{Objective: } \max_{\mathbf{q}} \sum_{i=1}^{L} \left( \frac{q_i}{L-i+1} \sum_{j=i}^{L} \omega_j \right),$$

$$\text{Constraints: } \sum_{i}^{L} S^{(q_i)} \leq \mathcal{B}. \tag{5}$$

where $S^{(q_i)}$ denotes the model size of the $i$-th layer under sparsity $q_i$, and $\mathcal{B}$ represents the target model size. The optimal sparsity configuration is given by $\mathbf{q}$. To maximize the model's representative capacity, our method try to assign smaller sparsity configurations to more independent layers by maximizing an objective function. For a more fine-grained sparsity allocation, we extend our approach to include intra-layer component-level sparsity allocation. After determining the sparsity ratios for each layer, we treat the remaining parameters in this layer as the target size and construct objective functions for its individual components. By applying ALS at this granular level, we obtain a secondary sparsity allocation, resulting in unique sparsity ratios for every layer and component. This hierarchical approach enables a highly customized and adaptable sparsity distribution throughout the entire network architecture, potentially leading to enhanced efficiency and performance gains.

## 4 Experimental Results

**Setup.** For pruning, we follow the settings of Wanda, SparseGPT, and Magnitude. Regarding the calibration data used in the linear optimization process, we follow the configurations of SparseGPT and Wanda, selecting data from the C4 dataset and ensuring that all test data are zero-shot. We use a calibration data size of 16 for linear optimization hyperparameters. The granularity, explained in Appendix. E.1 for linear optimization results is set to 0.5%. For the values of $x_i$, we use the input, although output and intermediate gates can also be used. Hyperparameter analysis is primarily conducted in the analysis section. Details about the experimental environment are provided in Appendix E.1.

**Evaluation and Metrics.** We measure the performance of pruned models through zero-shot tasks and language modeling. For zero-shot evaluation, we utilize seven tasks from the EleutherAI LM Harness [47]: Winogrande [39], PIQA [2], OpenBookQA [37], HellaSwag [61], BoolQ [6], ARC (Easy and Challenge) [7], and RTE (Recognizing Textual Entailment) [52]. We also include WikiText2 [36]. For the first seven datasets, we use the accuracy metric provided in the EleutherAI LM Harness. For WikiText2, we use the word_perplexity (PPL) metric. During evaluation, we ensure using the same database version, GPU model, and random seed.

**Models.** We evaluate the performance of ALS on LLMs, including LLaMA-V1 7B/13B/30B/65B [49], LLaMA-V2 7B/13B/70B [50], LLaMA-V3 8B [1], OPT 6.7B/13B [62].

Table 1: WikiText-2 perplexity performance of ALS at varying sparsity rates for sparse LLaMA-V2-7B/13B pruned by the Magnitude, SparseGPT, Wanda metric.

| Models | LLaMA-V2-7B | | | | | | LLaMA-V2-13B | | | | | |
|---|---|---|---|---|---|---|---|---|---|---|---|---|
| Sparse | 20% | 30% | 40% | 50% | 60% | 70% | 20% | 30% | 40% | 50% | 60% | 70% |
| Magnitude | 9.20 | 10.21 | 13.51 | 32.87 | 7.6e4 | 9e5 | **7.84** | 8.19 | 9.21 | 11.59 | 23.43 | 1.4e3 |
| *Magnitude w. ALS* | **8.91** | **9.60** | **11.03** | **15.19** | **83.23** | **2.8e5** | 7.86 | **8.18** | **8.95** | **10.78** | **18.52** | **204.17** |
| SparseGPT | 8.94 | 9.19 | 9.70 | 17.21 | 15.58 | 42.87 | **7.86** | **8.07** | **8.46** | 11.32 | **12.29** | 27.12 |
| *SparseGPT w. ALS* | **8.90** | **9.11** | **9.67** | **10.99** | **15.35** | **39.09** | 7.87 | 8.09 | 8.50 | **9.44** | 12.29 | **26.70** |
| Wanda | 8.92 | 9.23 | 9.85 | 12.31 | **19.57** | 219.46 | **7.88** | **8.11** | **8.56** | 11.21 | **14.42** | 116.99 |
| *Wanda w. ALS* | **8.90** | **9.15** | **9.81** | **11.61** | 20.91 | **214.10** | 7.90 | 8.14 | 8.63 | **9.86** | 14.81 | **91.58** |

Table 2: WikiText-2 perplexity performance of ALS at 50% sparsity rates for sparse LLaMA-V1-7B/13B/30B/65B, LLaMA-V2-7B/13B/70B, LLaMA-V3 8B/70B and OPT-6.7B/13B pruned by the Magnitude, SparseGPT, Wanda metric.

| Models | LLaMA-V1 | | | | LLaMA-V2 | | | LLaMA-V3 | | OPT | |
|---|---|---|---|---|---|---|---|---|---|---|---|
| Method | 7B | 13B | 30B | 65B | 7B | 13B | 70B | 8B | 70B | 6.7B | 13B |
| Dense | 9.38 | 8.20 | 6.09 | 4.93 | 8.71 | 7.68 | 4.52 | 7.26 | 2.92 | 11.29 | 11.33 |
| Magnitude | 42.26 | 43.61 | 13.68 | 8.88 | 32.87 | 11.59 | 8.59 | 1.1e3 | 19.29 | 1e3 | 4.1e4 |
| *Magnitude w. ALS* | **16.80** | **12.61** | **11.35** | **8.50** | **15.19** | **10.78** | **6.98** | **30.20** | **13.21** | **9.5e2** | **4e3** |
| SparseGPT | 18.35 | **9.90** | 10.07 | **7.82** | 17.21 | 11.32 | 7.84 | 16.87 | 8.49 | 13.35 | 12.53 |
| *SparseGPT w. ALS* | **11.87** | 9.97 | **8.28** | 7.97 | **10.99** | **9.44** | **7.58** | **10.23** | **7.24** | **12.29** | **11.49** |
| Wanda | 13.30 | 10.90 | 8.74 | 7.37 | 12.31 | 11.21 | 6.51 | 15.01 | 7.01 | 20.97 | 18.13 |
| *Wanda w. ALS* | **12.47** | **10.40** | **8.42** | **7.15** | **11.61** | **9.86** | **6.36** | **12.30** | **6.82** | **19.16** | **16.79** |

**Baselines.** We run ALS on LLMs with various methods, including Wanda [45], Magnitude-based pruning [18] and SparseGPT [16].

## 4.1 Language Modeling

**Quantitative Evaluation.** In Table 2, we compare the wikitext2 (PPL) performance of different pruning methods under 50% sparsity on the LLaMA-V1, LLaMA-V2, LLaMA-V3, and OPT models, including Dense (unpruned), Magnitude pruning [28], SparseGPT pruning [17], Wanda pruning [46], and the results of these pruning methods enhanced by ALS. The results show that the ALS generally improves the performance of various pruning methods.

For LLaMA-V1 models, Magnitude pruning shows high perplexity, e.g., 42.26 for the 7B model, reduced to 16.80 with ALS. SparseGPT performs better, with 18.35 for the 13B model, reduced to 11.87 with ALS. Wanda achieves the best results, with 13.30 for the 13B model, reduced to 12.47 with ALS.

For LLaMA-V2 and LLaMA-V3 models, ALS also reduces perplexity significantly. For instance, the Magnitude pruning in 13B LLaMA-V2 model drops from 15.19 to 10.78, and Wanda pruning in the 8B LLaMA-V3 model from 15.01 to 12.30 with ALS.

On the OPT model, perplexity significantly increases after pruning. For instance, the 13B model of OPT has a perplexity as high as 4.09e4 after Magnitude pruning, which remarkably reduces to 3.96e3 with ALS, demonstrating the effect of ALS in handling LLM pruning. However, there is an example where performance does not significantly improve with ALS. For instance, the 13B model of LLaMA-V1 has 9.90 perplexity after SparseGPT pruning, which slightly increases with ALS.

In summary, ALS significantly enhances model performance across various pruning methods by effectively mitigating performance loss.

**Varying Sparsity Rates.** Table 1 presents the perplexity scores of sparse LLaMA-V2 7B and 13B models pruned by Magnitude, SparseGPT, and Wanda methods, with and without ALS, at varying sparsity levels (20% to 70%). The results show that as sparsity increases, perplexity scores generally deteriorate, indicating a decline in language modeling performance. However, as the sparsity level increases, the performance gap between ALS and non-ALS methods widens, with ALS exhibiting better performance at most sparsity levels. This suggests that ALS can help mitigate the performance degradation caused by higher sparsity, becoming increasingly effective at maintaining LLMs performance as the sparsity level grows.

Table 3: Averaged accuracies (%) for zero-shot tasks at 50% sparsity rate for sparse LLaMA-V1 7B/13B/30B/65B, LLaMA-V2 7B/13B/70B, LLaMA-V3 8B and OPT-6.7B/13B .

| Models | LLaMA-V1 | | | | LLaMA-V2 | | | LLaMA-V3 | | OPT | |
|---|---|---|---|---|---|---|---|---|---|---|---|
| Method | 7B | 13B | 30B | 65B | 7B | 13B | 70B | 8B | 70B | 6.7B | 13B |
| Dense | 66.18 | 68.50 | 71.36 | 72.59 | 66.21 | 68.76 | 72.92 | 69.81 | 75.43 | 58.13 | 59.71 |
| Magnitude | 53.40 | 53.73 | 60.40 | 68.68 | 57.06 | 59.85 | 66.73 | 43.29 | 51.28 | 41.06 | 38.37 |
| *Magnitude w. ALS* | **56.28** | **61.21** | **62.61** | **69.42** | **60.09** | **63.39** | **70.28** | **57.43** | **53.64** | **43.31** | **40.92** |
| SparseGPT | 56.10 | **64.26** | 65.70 | 68.93 | 56.44 | 60.16 | 69.44 | 54.18 | 70.26 | 55.19 | 56.56 |
| *SparseGPT w. ALS* | **60.58** | 63.99 | **69.02** | **69.02** | **61.36** | **65.85** | **70.16** | **64.00** | **71.12** | **57.85** | **59.01** |
| Wanda | 58.87 | 64.74 | 68.54 | **71.71** | 61.88 | 64.48 | **71.87** | 58.12 | 72.25 | 47.81 | 50.46 |
| *Wanda w. ALS* | **61.47** | **64.82** | **69.35** | 71.41 | **62.84** | **66.58** | 71.75 | **62.48** | **73.12** | **47.89** | **50.50** |

Table 4: Accuracies (%) for zero-shot tasks with 50% sparsity using LLaMA-V2 family.

| LLaMA-V2 | Method | winogrande | piqa | openbookqa | hellaswag | boolq | arc_easy | arc_challenge | rte | Mean |
|---|---|---|---|---|---|---|---|---|---|---|
| | Dense | 69.06 | 79.11 | 44.20 | 75.98 | 77.74 | 74.49 | 46.25 | 62.82 | 66.21 |
| | Magnitude | 63.30 | 73.67 | 38.80 | 65.58 | 62.94 | 57.70 | 37.46 | **57.04** | 57.06 |
| | *Magnitude w. ALS* | **65.19** | **75.46** | **41.40** | **69.11** | **71.38** | **63.47** | **39.51** | 55.24 | **60.09** |
| 7B | SparseGPT | 63.14 | 71.71 | 35.60 | 63.91 | 69.05 | 58.29 | 34.98 | 54.87 | 56.44 |
| | *SparseGPT w. ALS* | **67.96** | **76.39** | **40.00** | **70.52** | **70.98** | **67.97** | **41.13** | **55.96** | **61.36** |
| | Wanda | 67.32 | 76.99 | 41.40 | 68.76 | **75.78** | 69.23 | 41.72 | 53.83 | 61.88 |
| | *Wanda w. ALS* | **67.80** | **77.10** | **44.80** | **70.75** | 75.47 | **69.61** | **42.32** | 54.87 | **62.84** |
| | Dense | 72.38 | 80.52 | 45.20 | 79.39 | 80.58 | 77.53 | 49.15 | 65.34 | 68.76 |
| | Magnitude | 65.27 | 77.20 | 40.60 | 73.01 | 57.68 | 67.17 | 41.89 | **55.96** | 59.85 |
| | *Magnitude w. ALS* | **68.35** | **77.48** | **43.60** | **74.42** | **73.15** | **69.91** | **44.63** | 55.60 | **63.39** |
| 13B | SparseGPT | 67.25 | 75.84 | 41.20 | 69.63 | 68.50 | 63.76 | 38.40 | 56.68 | 60.16 |
| | *SparseGPT w. ALS* | **71.19** | **78.13** | **44.40** | **74.99** | **81.16** | **69.82** | **43.94** | **63.18** | **65.85** |
| | Wanda | 69.39 | 78.13 | 44.10 | 75.02 | 80.34 | **70.37** | 42.76 | 55.72 | 64.48 |
| | *Wanda w. ALS* | **72.06** | **78.51** | **45.80** | **75.67** | **81.35** | 70.33 | **46.08** | **62.82** | **66.58** |
| | Dense | 77.98 | 82.70 | 48.80 | 83.80 | 83.79 | 81.06 | 57.34 | 67.87 | 72.92 |
| | Magnitude | 73.64 | 79.54 | 44.20 | 79.29 | 71.07 | 74.79 | 50.68 | 60.65 | 66.73 |
| | *Magnitude w. ALS* | **74.74** | **80.96** | **46.40** | **80.57** | **79.63** | **77.99** | **54.10** | **67.87** | **70.28** |
| 70B | SparseGPT | 75.37 | 79.38 | 44.80 | 79.88 | 82.81 | 77.02 | 48.38 | 67.87 | 69.44 |
| | *SparseGPT w. ALS* | **76.72** | **80.36** | **45.20** | **80.09** | **81.56** | **77.86** | **50.17** | **69.31** | **70.16** |
| | Wanda | 77.58 | 81.18 | **46.20** | 80.95 | **83.94** | 78.91 | 54.69 | **71.48** | **71.87** |
| | *Wanda w. ALS* | 77.27 | **81.61** | 45.80 | **81.30** | 82.54 | 78.54 | **55.80** | 71.12 | 71.75 |

## 4.2 Zero-shot Tasks

In Table 3, we present the averaged accuracy performance of pruned LLaMA-V1, LLaMA-V2, LLaMA-V3, and OPT models on seven downstream zero-shot tasks at a 50% sparsity ratio. For detailed performance on specific tasks, please refer to Table 4, which shows improvements in most tasks. The average accuracy across the majority of tasks demonstrates the effectiveness of ALS in enhancing sparse large language models of any scale. Remarkably, for LLaMA3-8B, the incorporation of ALS leads to an improvement of 14.14% and 9.87% compared to the Magnitude and SparseGPT baselines, respectively. Similarly, for LLaMA1-13B, the addition of ALS results in an improvement of 5.95% compared to the baselines.

The significant performance improvement of LLaMA-V3 8B may be attributed to the fact that the new model is not well-suited for uniform pruning methods. we further use the LLaMA-V3 8B model as an example to intuitively present the improvements brought by ALS from the perspective of the heat map (Fig. 3 in the Appendix D). The heat map reveals that, at 50% sparsity, the redundancy between layers of the LLaMA-V3 8B model exhibits a distinct pattern compared to other models. The green distribution on both sides indicates that there is small redundancy between the shallow layers and other layers. This suggests that the information captured by the middle and deep layers has more overlap and similarity with other layers. The ALS method takes advantage of this inter-layer redundancy pattern and apply high sparsity ratio into the layers with higher redundancy. It maximally preserves the key information and reduces the impact of sparsification on model performance. From the heat maps of different models in the Appendix D, it can also be observed that each model requires a different sparsity ratio because the redundancy between its layers varies.

Table 5: WikiText-2 perplexity of Wanda with ALS at 50% sparsity on LLaMA-family models.

| Models | V1-7B | V1-13B | V1-30B | V2-7B | V2-13B |
|---|---|---|---|---|---|
| Dense | 9.38 | 8.20 | 6.09 | 8.71 | 7.68 |
| *Wanda* | 13.30 | 10.90 | 8.74 | 12.31 | 11.21 |
| *w. ALS* | 12.47 | 10.40 | 8.42 | 11.61 | 9.86 |
| *w. LoRA* | 7.65 | 9.25 | 6.99 | 9.89 | 8.30 |

Table 6: Zero-shot accuracy (%) of Wanda with ALS at 50% sparsity on LLaMA-family models.

| Models | V1-7B | V1-13B | V1-30B | V2-7B | V2-13B |
|---|---|---|---|---|---|
| Dense | 66.18 | 68.50 | 71.36 | 66.21 | 68.76 |
| *Wanda* | 58.87 | 64.74 | 68.54 | 61.88 | 64.48 |
| *w. ALS* | 61.47 | 64.82 | 69.35 | 62.84 | 66.58 |
| *w. LoRA* | 71.64 | 66.71 | 71.44 | 65.05 | 67.81 |

Table 7: Results of feature choice from varying component output of each layer on WikiText2 and zero-shot tasks.

| Feature Choice | PPL | ACC |
|---|---|---|
| In | 10.070 | 60.75 |
| Out | 10.012 | 60.56 |
| Gate | 10.030 | 60.76 |

Table 8: Impact of normalizing features and per-layer weights for distance function on WikiText2 and zero-shot tasks .

| Distance Function | PPL | ACC |
|---|---|---|
| Vanilla | 10.078 | 66.40 |
| Feature-Norm | 10.076 | 66.40 |
| Feature-Norm+Weight-Norm | 10.070 | 68.11 |

## 4.3 Ablation Study

In this part, we examine the impact of various components within the ALS framework and compare it with LoRA Fine-tuning, specifically focusing on its bound setting, standardization on weight or feature, granularity choice, which can be found in Appendix. E.1, feature choice and robustness to calibration samples. All experimental setups are based on the LLaMA2-13B model with Wanda pruning and ALS.

**Comparison with LoRA Fine-tuning.** Our experiments in Table 5 and Table 6 demonstrate the substantial benefits of combining Wanda+ALS with LoRA fine-tuning across the LLaMA model family. The improvements are most striking in the LLaMA-V1 7B model, which showed a dramatic reduction in perplexity by 4.82 alongside a 10.17% increase in accuracy. Larger V1 models also benefited, with the 30B and 13B variants showing perplexity reductions of 1.43 and 1.15, coupled with accuracy gains of 2.09% and 1.89% respectively. The LLaMA-V2 models exhibited similar positive trends, with both 7B and 13B versions showing perplexity improvements and accuracy increases. These impressive results were achieved using just 2000 C4 samples for LoRA fine-tuning in a zero-shot setting, highlighting the method's efficiency and effectiveness even with limited training data unrelated to the evaluation tasks.

**Feature Selection and Normalization.** Table 7 (a) compares the performance of input, output, and gate features in capturing layer independence, with output features achieving slightly lower perplexity. Table 8 (b) demonstrates the significant impact of jointly normalizing features and per-layer weights. Applying this normalization strategy yields a substantial improvement in accuracy, increasing from 66.40% to 68.11%, while also reducing perplexity from 10.078 to 10.070.

**Comparison with OWL.** We compared the performance of our proposed method with the OWL method on a set of benchmark datasets. The results are summarized in Table 9. We adopted the optimal parameter settings described in the OWL paper. Across all tested configurations, our method consistently achieved lower values compared to OWL, demonstrating its superior performance. Specifically, in the unstructured 50% setting, Wanda with ALS outperformed OWL by a margin of 0.25 units. Furthermore, in the structured pruning settings of 2:4 and 4:8, the advantage of Wanda with ALS increased to 0.95 and 0.44, respectively.

**N:M Results.** We also investigated the performance of our method in the N:M setting, where $N$ features are selected from $M$ available features. The results are shown in Table 9. Similarly, for OWL, we used the optimal parameter combination reported in their paper. Across all N:M configurations, ALS consistently achieved lower values compared to OWL. As the number of selected features $N$ increased, both methods exhibited performance improvements, but the advantage of ALS became more pronounced. For instance, in the 2:4 case, ALS outperformed OWL by a margin of 0.95 units, and this gap further widened to 1.82 in the 4:8 case. Overall, OWL is a method that is highly sensitive to parameter settings, and obtaining the optimal parameters may require dozens of experiments to determine the best combination. Moreover, there is no clear theoretical analysis explaining why such a combination should be used.

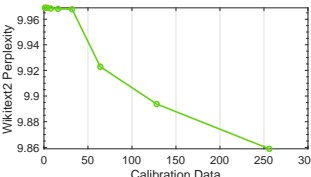 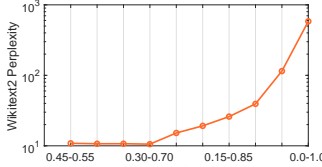 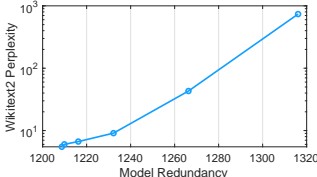

Figure 2: (a) Calibration data experiment: PPL decreases slightly with more data. (b) Pruning bounds: Model performance remains relatively stable between 30% and 70% bounds. (c) Model redundancy: Higher RM metric, lower performance.

Table 9: WikiText-2 perplexity performance on LLaMA-V2-13B at 50% sparsity rates.

| Ratio | Wanda w. ALS | Wanda w. OWL |
|---|---|---|
| 50% | **9.86** | 10.11 |
| 2:4 | **15.52** | 16.47 |
| 4:8 | **11.65** | 12.09 |

Table 10: Pruning speed of various methods with ALS on LLaMA-V2-7B.

| | ALS | | |
|---|---|---|---|
| Base Method | RM (s) | LP (ms) | Total (min) |
| Magnitude (1.62s) | 88.59 | 169 | 1.51 |
| SparseGPT (1058s) | 91.32 | 158 | 19.16 |
| Wanda (199s) | 89.47 | 160 | 4.81 |

**Calibration Data.** In Fig. 2 (a), we present the performance of pruning methods with different numbers of calibration samples. We use the size of 2, 4, 8, 16, 32, 64, 128, 256. Although this experiment reveals that the model's performance improves with an increase in the size of the calibration data, the improvement is quite limited. Even when comparing the scales of 2 and 256 in calibration samples, the perplexity decreases by only 0.11. These results further highlight the robustness of ALS.

**Boundes.** In Fig. 2 (b) demonstrates the effect of pruning bounds on the performance of the LLaMA-V2 13B model. When the pruning bounds are set too high (e.g., 0.0-1.0), the model's performance significantly deteriorates from $10^1 to 10^3$ compared with 0.3-0.7, indicating that aggressive pruning may impair the model's representational capacity. However, when the pruning bounds are set between 30% and 70%, the model's performance remains nearly unaffected.

**Computation efficiency.** As shown in Table 10, our ALS involves two computational phases: the Redundancy Metric (RM) calculation, which consistently takes approximately 90 seconds across all methods, and the Linear Programming (LP) solution, requiring roughly 160-170 milliseconds. The total processing time varies notably depending on the base pruning method employed: Magnitude pruning, requiring just 1.62 seconds for its base operation, achieves the fastest total completion time of 1.51 minutes when combined with ALS. Wanda, with its base pruning time of 199 seconds, completes the entire process in 4.81 minutes, while SparseGPT, requiring 1058 seconds for its base operation, takes 19.16 minutes in total. Compared to BESA [57] with 4.5 hours for sparsity allocation and pruning, our approach is notably faster, completing the process in minutes rather than hours.

## 5 Conclusion

In this work, we present Adaptive Layer Sparsity (ALS), a novel approach for optimizing LLMs through the efficient allocation of sparsity across layers. By minimizing inter-layer redundancy, ALS achieves significant model compression while maintaining performance, as demonstrated through extensive experiments on diverse LLMs and tasks. We hope ALS offers valuable insights and practical tools for deploying LLMs under limited computational resources, and that our work may shed light on the role of sparsity in LLMs and its potential for model optimization. Future research will explore the relationship between sparsity allocation and individual weight importance, and investigate the integration of dynamic sparsity allocation with pruning metrics. By pushing the boundaries of model compression and efficiency, we aim to enhance the development of more capable and accessible LLMs for diverse applications.

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

## Appendix

## A   Limitation: Computational Complexity of Intra-Layer Component Independence Calculation

In the ALS method, a key step is computing the independence matrix between layers. While calculating the matrix at the layer level for a 70 billion parameter model with 80 layers requires only an $80 \times 80$ RM matrix, including intra-layer components increases the matrix size to $7 \times 80 \times 7 \times 80$, leading to 49 times more computational time and resources.

- **High Computational Complexity**: The increased matrix size results in exponential growth in computation and resource consumption.
- **Excessive Memory Usage**: High-dimensional matrix computations require substantial memory, potentially exceeding hardware capacities.

**Solutions**

- **Hybrid Solution with C++**: Store intermediate data locally and use C++ to handle calculations and solve the linear programming problem. This approach can be up to 50 times faster than using Python alone.
- Alternatively, calculate sparsity ratios for each layer, determine the parameters to retain, and use these as the target size for further linear programming. This approach requires only an $80 \times 80$ RM matrix and 7 additional $7 \times 7$ matrices, without significantly increasing computation time.

Hybrid Solution with C++ is a preferred solution because it will keep most of independent components to maintain the model performance after pruning.

## B   Detailed Formulae Derivation

This section provides a detailed derivation of the key mathematical formulae used in the paper.

### B.1   Detailed Derivation of the Mutual Information Approximation

The objective of LLMs is to minimize the reconstruction error between the outputs of the sparse and dense models. We start by defining the mutual information between the outputs of different layers to quantify the redundancy within the model. Mutual information $I(X;Y)$ measures the amount of information obtained about one random variable $X$ through observing the other random variable $Y$. It quantifies the reduction in uncertainty of $X$ due to the knowledge of $Y$. This measure helps identify layers with high redundancy, which can be pruned to achieve a more efficient representation.

The mutual information $I(X_i; X_j)$ between two layers $i$ and $j$ is given by:

$$I(X_i; X_j) = H(X_i) + H(X_j) - H(X_i, X_j) \tag{6}$$

where $H(X_i)$ and $H(X_j)$ are the marginal entropies, and $H(X_i, X_j)$ is the joint entropy. The entropy $H(X)$ measures the average amount of information or uncertainty in a random variable $X$. These entropies are defined as:

$$H(X_i) = -\int p(x_i) \log p(x_i) \, \mathrm{d}x_i \tag{7}$$

$$H(X_j) = -\int p(x_j) \log p(x_j) \, \mathrm{d}x_j \tag{8}$$

$$H(X_i, X_j) = -\int p(x_i, x_j) \log p(x_i, x_j) \, \mathrm{d}x_i \, \mathrm{d}x_j \tag{9}$$

The mutual information can also be expressed in terms of the probability density functions as:

$$I(X_i; X_j) = \int p(x_i, x_j) \log \frac{p(x_i, x_j)}{p(x_i) p(x_j)} \mathrm{d}x_i \, \mathrm{d}x_j \tag{10}$$

To approximate these values, we use Monte Carlo sampling. Given $N$ i.i.d. samples $\mathbf{x}^{(1)}, \mathbf{x}^{(2)}, \ldots, \mathbf{x}^{(N)}$ drawn from the joint distribution $p(x_i, x_j)$, the mutual information can be approximated as:

$$\hat{I}(X_i; X_j) \approx \frac{1}{N} \sum_{n=1}^{N} \log \frac{p\left(x_i^{(n)}, x_j^{(n)}\right)}{p\left(x_i^{(n)}\right) p\left(x_j^{(n)}\right)} \tag{11}$$

Using kernel density estimation to approximate the probability densities. For instance:

$$\hat{p}(\mathbf{y}) = \frac{1}{Nh^d} \sum_{i=1}^{N} K\left(\frac{\mathbf{y} - \mathbf{y}_i}{h}\right) \tag{12}$$

Here, $K(\cdot)$ is the kernel function, $h$ is the bandwidth parameter, and $d$ is the dimension of $\mathbf{y}$. Kernel density estimation is a non-parametric method for estimating the probability density function $p(\mathbf{y})$ of a random vector $\mathbf{y}$, given a set of i.i.d. samples $\mathbf{Y} = \{\mathbf{y}_1, \mathbf{y}_2, \ldots, \mathbf{y}_N\}$ drawn from $p(\mathbf{y})$. Commonly used kernels include the Gaussian and Epanechnikov kernels.

To compute the marginal and joint probability density functions of the samples' outputs at the $i$-th and $j$-th layers, we apply kernel density estimation:

$$\hat{p}(\mathbf{x}_i) = \frac{1}{Nh_i^{d_i}} \sum_{n=1}^{N} K\left(\frac{\mathbf{x}_i - \mathbf{x}_i^{(n)}}{h_i}\right) \tag{13}$$

$$\hat{p}(\mathbf{x}_j) = \frac{1}{Nh_j^{d_j}} \sum_{n=1}^{N} K\left(\frac{\mathbf{x}_j - \mathbf{x}_j^{(n)}}{h_j}\right) \tag{14}$$

$$\hat{p}(\mathbf{x}_i, \mathbf{x}_j) = \frac{1}{Nh_i^{d_i} h_j^{d_j}} \sum_{n=1}^{N} K\left(\frac{\mathbf{x}_i - \mathbf{x}_i^{(n)}}{h_i}\right) K\left(\frac{\mathbf{x}_j - \mathbf{x}_j^{(n)}}{h_j}\right) \tag{15}$$

where $d_i$ and $d_j$ are the dimensions of $\mathbf{x}_i$ and $\mathbf{x}_j$, respectively.

To derive the mutual information approximation, let's start with the Monte Carlo approximation of mutual information (Equation 7). Substituting the kernel density estimates (Equations 9-11) into Equation 7, we get:

$$\hat{I}(X_i; X_j) \approx \frac{1}{N} \sum_{n=1}^{N} \log \frac{\sum_{k=1}^{N} K\left(\frac{\mathbf{x}_i^{(n)} - \mathbf{x}_i^{(k)}}{h_i}\right) K\left(\frac{\mathbf{x}_j^{(n)} - \mathbf{x}_j^{(k)}}{h_j}\right)}{\left(\sum_{k=1}^{N} K\left(\frac{\mathbf{x}_i^{(n)} - \mathbf{x}_i^{(k)}}{h_i}\right)\right) \left(\sum_{k=1}^{N} K\left(\frac{\mathbf{x}_j^{(n)} - \mathbf{x}_j^{(k)}}{h_j}\right)\right)} \tag{16}$$

Now, consider the feature matrix inner product approximation for the kernel function. Let:

$$K\left(\frac{\mathbf{x}_i^{(n)} - \mathbf{x}_i^{(k)}}{h_i}\right) \approx \left\|\mathbf{x}_i^{(n)T} \mathbf{x}_i^{(k)}\right\|_F \tag{17}$$

$$K\left(\frac{\mathbf{x}_j^{(n)} - \mathbf{x}_j^{(k)}}{h_j}\right) \approx \left\|\mathbf{x}_j^{(n)T} \mathbf{x}_j^{(k)}\right\|_F \tag{18}$$

where $\|\cdot\|_F$ denotes the Frobenius norm.

Therefore, the joint kernel function can be approximated by:

$$K\left(\left(\mathbf{x}_i^{(n)}, \mathbf{x}_j^{(n)}\right), \left(\mathbf{x}_i^{(k)}, \mathbf{x}_j^{(k)}\right)\right) \approx \left\|\mathbf{x}_i^{(n)T} \mathbf{x}_j^{(n)}\right\|_F \tag{19}$$

Substituting these approximations (Equations 13-15) back into the mutual information formula (Equation 12), we get:

$$\hat{I}(X_i; X_j) \approx \frac{1}{N} \sum_{n=1}^{N} \log \frac{\left\|\mathbf{x}_i^{(n)T} \mathbf{x}_j^{(n)}\right\|_F}{\left\|\mathbf{x}_i^{(n)T} \mathbf{x}_i^{(n)}\right\|_F \left\|\mathbf{x}_j^{(n)T} \mathbf{x}_j^{(n)}\right\|_F} \tag{20}$$

This leads to the final expression for the mutual information approximation using Monte Carlo sampling and kernel density estimation with the feature matrix inner product approximation.

## B.2 Redundancy Metric Derivation

Since mutual information has no general upper bound, its upper limit depends on the entropy of either $X_i$ or $X_j$, which represent the $i$-th and $j$-th layer output of a batch input. To address this, we can use functions to transform the range of mutual information, ensuring a bounded and more interpretable metric. For instance, using $e^{\hat{I}(X_i;X_j)}$ or a Gaussian function, we can redefine the measure as follows. Considering $\left\|\mathbf{x}_i^T \mathbf{x}_j\right\|_F e^{\hat{I}(X_i;X_j)}$ for a batch of inputs, we can remove the sum and $\frac{1}{N}$, then obtain:

$$\left\|\mathbf{X}_i^T \mathbf{X}_j\right\|_F \cdot e^{\hat{I}(X_i;X_j)} = \frac{\left\|\mathbf{X}_i^T \mathbf{X}_j\right\|_F^2}{\left\|\mathbf{X}_i^T \mathbf{X}_i\right\|_F \left\|\mathbf{X}_j^T \mathbf{X}_j\right\|_F} \tag{21}$$

To derive the Redundancy Metric (RM) formula, we consider that a batch is a set of samples processed simultaneously by the neural network. Because a batch is input simultaneously, we remove the summation and division by $N$, obtaining the following formula:

$$RM\left(\mathbf{X}_i, \mathbf{X}_j\right) = \frac{\left\|\mathbf{X}_i^T \mathbf{X}_j\right\|_F^2}{\left\|\mathbf{X}_i^T \mathbf{X}_i\right\|_F \left\|\mathbf{X}_j^T \mathbf{X}_j\right\|_F} \tag{22}$$

In this formula, $RM(\cdot) \in [0,1]$, where a value close to 0 indicates high independence between the layers (low redundancy), and a value close to 1 indicates high redundancy (low independence).

## B.3 Constraints and Objective Functions

Our redundancy metric is used to guide the allocation of sparsity ratios across layers. We construct a redundancy matrix $\mathbf{\Psi} \in \mathbb{R}^{L \times L}$ where each element $\psi_{ij}$ represents the redundancy between layers $i$ and $j$, computed using Equation 18. The overall redundancy of each layer is:

$$\rho_i = \sum_{j=1}^{L} \psi_{ij} - 1 \tag{23}$$

A lower $\rho_i$ indicates higher independence. Using a monotonically decreasing function, we define the importance factor $\omega_i$ as:

$$\omega_i = e^{-\frac{1}{\mu}\rho_i} \tag{24}$$

where $\mu$ is a hyperparameter controlling the decay rate.

Our objective is to maximize the sum of the weighted sparsity ratios across all layers, subject to a model size constraint $\mathcal{B}$. The sparsity ratio $q_i \in [0,1]$ represents the fraction of weights to be pruned in the $i$-th layer. The optimization problem can be formulated as:

$$\text{maximize}_{\mathbf{q}} \quad \sum_{i=1}^{L} \left( \frac{q_i}{L-i+1} \sum_{j=i}^{L} \omega_j \right) \tag{25}$$

$$\text{subject to} \quad \sum_{i=1}^{L} S^{(q_i)} \leq \mathcal{B} \tag{26}$$

$$0 \leq q_i \leq 1, \quad \forall i \in \{1, 2, \ldots, L\} \tag{27}$$

where $S^{(q_i)}$ represents the model size of the $i$-th layer under sparsity ratio $q_i$, and $\mathbf{q} = [q_1, q_2, \ldots, q_L]^T$ is the vector of sparsity ratios for all $L$ layers. The objective function (Equation 22) aims to allocate higher sparsity ratios to layers with higher importance factors. The constraint (Equation 23) ensures that the total model size after pruning does not exceed the budget $\mathcal{B}$.

# C More Experimental Results

## C.1 Zero-Shot results Details

**Results in LLaMA-V1,V2,V3 and OPT series at 50% Sparsity**

Table 11: Accuracies (%) for zero-shot tasks with 50% sparsity using LLaMA-V1 family.

| Llama V1 | Method | winogrande | piqa | openbookqa | hellaswag | boolq | arc_easy | arc_challenge | rte | Mean |
|---|---|---|---|---|---|---|---|---|---|---|
| 7B | Dense | 70.09 | 79.16 | 44.40 | 76.18 | 75.11 | 72.98 | 44.71 | 66.79 | 66.18 |
| | Magnitude | 59.35 | 71.38 | 35.00 | 60.89 | 54.56 | 54.38 | 37.12 | 54.51 | 53.40 |
| | *Magnitude w. ALS* | **61.25** | **74.16** | **36.60** | **65.62** | **59.82** | **59.98** | **38.31** | 54.51 | **56.28** |
| | SparseGPT | 63.06 | 73.61 | 37.40 | 64.62 | 64.43 | 55.68 | 36.60 | 53.43 | 56.10 |
| | *SparseGPT w. ALS* | **66.77** | **76.33** | **39.00** | **68.83** | **74.28** | **64.84** | **39.59** | **54.87** | **60.19** |
| | Wanda | 66.38 | 74.76 | **39.00** | 68.92 | 70.70 | 61.74 | 38.91 | 50.54 | 58.87 |
| | *SparseGPT w. ALS* | 66.30 | **77.26** | 38.60 | **69.59** | 73.70 | **65.66** | **40.02** | **60.65** | **61.47** |
| 13B | Dense | 72.77 | 80.14 | 44.80 | 79.06 | 77.89 | 74.79 | 47.78 | 70.76 | 68.50 |
| | Magnitude | 63.38 | 70.95 | 39.80 | 59.69 | 54.95 | 54.29 | 35.84 | 50.90 | 53.73 |
| | *Magnitude w. ALS* | **68.04** | **76.39** | **42.00** | **71.10** | **70.70** | **63.81** | **40.27** | **57.40** | **61.21** |
| | SparseGPT | 70.96 | 76.66 | **45.20** | 73.72 | 74.89 | 67.47 | 42.75 | 62.45 | 64.26 |
| | *SparseGPT w. ALS* | **71.35** | **77.31** | 43.60 | 73.59 | 74.19 | **67.64** | 42.49 | 61.73 | 63.99 |
| | Wanda | **71.51** | **77.91** | 43.60 | 74.13 | **75.96** | 69.65 | 43.77 | 61.37 | 64.74 |
| | *SparseGPT w. ALS* | 71.35 | 77.37 | 43.00 | **74.34** | 75.17 | **69.70** | **44.45** | **63.18** | **64.82** |
| 30B | Dense | 75.85 | 82.26 | 48.40 | 82.63 | 82.72 | 78.96 | 52.90 | 67.15 | 71.36 |
| | Magnitude | 66.54 | 75.68 | 41.20 | 67.28 | 64.31 | 70.75 | **47.27** | 50.18 | 60.40 |
| | *Magnitude w. ALS* | **69.93** | **77.04** | **41.60** | **68.67** | **74.19** | **72.52** | 46.76 | 50.18 | **62.61** |
| | SparseGPT | 71.03 | 77.80 | 42.40 | 76.06 | 77.89 | 73.04 | 47.44 | 59.57 | 65.92 |
| | *SparseGPT w. ALS* | **74.51** | **78.78** | **45.40** | 78.54 | 80.43 | 77.61 | **52.40** | **64.62** | **69.02** |
| | Wanda | **74.51** | 79.60 | **47.40** | 79.31 | 80.64 | 77.57 | 51.54 | 57.76 | 68.54 |
| | *SparseGPT w. ALS* | 73.64 | **80.20** | 46.80 | **79.50** | **81.28** | **78.37** | **53.67** | 61.37 | **69.35** |
| 65B | Dense | 77.43 | 82.26 | 47.00 | 84.14 | 84.86 | 79.80 | 55.55 | 69.68 | 72.59 |
| | Magnitude | 74.66 | 79.43 | 48.00 | 79.90 | 79.63 | 73.65 | **51.71** | 62.45 | 68.68 |
| | *Magnitude w. ALS* | **74.98** | **80.36** | **48.40** | **79.91** | **80.92** | **74.62** | 50.85 | **65.34** | **69.42** |
| | SparseGPT | **74.98** | **81.01** | 44.40 | 80.19 | **83.46** | **75.55** | 48.63 | 63.18 | 68.93 |
| | *SparseGPT w. ALS* | 73.95 | 80.90 | **44.80** | **80.35** | 83.33 | 74.03 | **49.06** | **65.70** | **69.02** |
| | Wanda | **77.19** | 80.69 | **48.60** | 81.72 | **84.71** | 77.56 | 52.47 | 70.40 | **71.71** |
| | *Wanda w. ALS* | 76.40 | **81.39** | 47.00 | 81.68 | 84.68 | 76.94 | **52.39** | **70.76** | 71.41 |

Table 12: Accuracies (%) for zero-shot tasks with 50% sparsity using LLaMA-V2 family.

| Llama V2 | Method | winogrande | piqa | openbookqa | hellaswag | boolq | arc_easy | arc_challenge | rte | Mean |
|---|---|---|---|---|---|---|---|---|---|---|
| 7B | Dense | 69.06 | 79.11 | 44.20 | 75.98 | 77.74 | 74.49 | 46.25 | 62.82 | 66.21 |
| | Magnitude | 63.30 | 73.67 | 38.80 | 65.58 | 62.94 | 57.70 | 37.46 | **57.04** | 57.06 |
| | *Magnitude w. ALS* | **65.19** | **75.46** | **41.40** | **69.11** | **71.38** | **63.47** | **39.51** | 55.24 | **60.09** |
| | SparseGPT | 63.14 | 71.71 | 35.60 | 63.91 | 69.05 | 58.29 | 34.98 | 54.87 | 56.44 |
| | *SparseGPT w. ALS* | **67.96** | **76.39** | **40.00** | **70.52** | **70.98** | **67.97** | **41.13** | **55.96** | **61.36** |
| | Wanda | 67.32 | 76.99 | 41.40 | 68.76 | **75.78** | 69.23 | 41.72 | 53.83 | 61.88 |
| | *SparseGPT w. ALS* | **67.80** | **77.10** | **44.80** | **70.75** | 75.47 | **69.61** | **42.32** | **54.87** | **62.84** |
| 13B | Dense | 72.38 | 80.52 | 45.20 | 79.39 | 80.58 | 77.53 | 49.15 | 65.34 | 68.76 |
| | Magnitude | 65.27 | 77.20 | 40.60 | 73.01 | 57.68 | 67.17 | 41.89 | **55.96** | 59.85 |
| | *Magnitude w. ALS* | **68.35** | **77.48** | **43.60** | **74.42** | **73.15** | **69.91** | **44.63** | 55.60 | **63.39** |
| | SparseGPT | 67.25 | 75.84 | 41.20 | 69.63 | 68.50 | 63.76 | 38.40 | 56.68 | 60.16 |
| | *SparseGPT w. ALS* | **71.19** | **78.13** | 44.40 | **74.99** | **81.16** | **69.82** | **43.94** | **63.18** | **65.85** |
| | Wanda | 69.39 | 78.13 | 44.10 | 75.02 | 80.34 | **70.37** | 42.76 | 55.72 | 64.48 |
| | *SparseGPT w. ALS* | **72.06** | **78.51** | **45.80** | **75.67** | **81.35** | 70.33 | **46.08** | **62.82** | **66.58** |
| 70B | Dense | 77.98 | 82.70 | 48.80 | 83.80 | 83.79 | 81.06 | 57.34 | 67.87 | 72.92 |
| | Magnitude | 73.64 | 79.54 | 44.20 | 79.29 | 71.07 | 74.79 | 50.68 | 60.65 | 66.73 |
| | *Magnitude w. ALS* | **74.74** | **80.96** | **46.40** | **80.57** | **79.63** | **77.99** | **54.10** | **67.87** | **70.28** |
| | SparseGPT | 75.37 | 79.38 | 44.80 | 79.88 | 82.81 | 77.02 | 48.38 | 67.87 | 69.44 |
| | *SparseGPT w. ALS* | **76.72** | **80.36** | **45.20** | **80.09** | 81.56 | **77.86** | **50.17** | **69.31** | **70.16** |
| | Wanda | **77.58** | 81.18 | **46.20** | 80.95 | **83.94** | 78.91 | 54.69 | **71.48** | **71.87** |
| | *SparseGPT w. ALS* | 77.27 | **81.61** | 45.80 | **81.30** | 82.54 | 78.54 | **55.80** | 71.12 | 71.75 |

Table 13: Accuracies (%) for zero-shot tasks with 50% sparsity using LLaMA-V3.

| Llama V3 | Method | winogrande | piqa | openbookqa | hellaswag | boolq | arc_easy | arc_challenge | rte | Mean |
|---|---|---|---|---|---|---|---|---|---|---|
| | Dense | 73.01 | 80.52 | 44.80 | 79.15 | 81.19 | 77.61 | 53.24 | 68.95 | 69.81 |
| 8B | Magnitude | 52.72 | 59.90 | 35.20 | 29.78 | 42.84 | 43.01 | 29.78 | 53.07 | 43.29 |
| | *Magnitude w. ALS* | **65.35** | **70.95** | **37.00** | **65.50** | **68.90** | **59.93** | **37.29** | **54.51** | **57.43** |
| | SparseGPT | 64.17 | 71.11 | 35.80 | 46.52 | 69.02 | 58.42 | 34.64 | 53.79 | 54.18 |
| | *SparseGPT w. ALS* | **70.72** | **75.14** | **41.00** | **71.14** | **80.52** | **70.08** | **45.31** | **58.12** | **64.00** |
| | Wanda | 66.93 | 73.39 | 37.20 | 47.03 | 75.90 | 65.66 | 39.25 | **59.57** | 58.12 |
| | *Wanda w. ALS* | **70.17** | **74.97** | **39.20** | **67.65** | **79.14** | **67.47** | **42.41** | 58.84 | **62.48** |

Table 14: Accuracies (%) for zero-shot tasks with 50% sparsity using OPT family.

| OPT | Method | winogrande | piqa | openbookqa | hellaswag | boolq | arc_easy | arc_challenge | rte | Mean |
|---|---|---|---|---|---|---|---|---|---|---|
| | Dense | 66.19 | 76.50 | 38.20 | 67.91 | 66.15 | 60.10 | 34.73 | 55.23 | 58.13 |
| | Magnitude | 50.67 | 63.17 | 29.60 | 32.41 | 38.01 | 38.09 | 23.81 | **52.71** | 41.06 |
| | ***Magnitude w. ALS*** | **52.41** | **64.69** | **30.00** | **39.85** | **40.86** | **41.46** | **26.28** | 50.90 | **43.31** |
| 6.7B | SparseGPT | 63.69 | 74.86 | 36.60 | 61.52 | 62.88 | 56.73 | 31.06 | 54.15 | 55.19 |
| | ***SparseGPT w. ALS*** | **65.27** | **76.77** | **37.40** | **67.17** | **65.84** | **60.23** | **34.56** | **55.60** | **57.85** |
| | Wanda | **58.17** | 64.04 | **29.60** | 47.17 | 60.86 | **46.17** | 25.94 | 50.54 | 47.81 |
| | ***SparseGPT w. ALS*** | 56.83 | **65.02** | 29.00 | **48.35** | **60.89** | 46.13 | 25.68 | **51.26** | **47.89** |
| | Dense | 67.88 | 76.77 | 39.00 | 69.84 | 68.87 | 61.87 | 35.67 | 57.76 | 59.71 |
| | Magnitude | 48.62 | 53.26 | 25.60 | **26.92** | 45.02 | 30.68 | 24.15 | **52.71** | 38.37 |
| | ***Magnitude w. ALS*** | **51.14** | **54.90** | 25.60 | 26.43 | **61.53** | **30.98** | **25.17** | 51.63 | **40.92** |
| 13B | SparseGPT | 63.62 | 74.27 | 38.40 | 64.32 | 62.36 | 59.43 | 34.47 | 55.60 | 56.56 |
| | ***SparseGPT w. ALS*** | **65.27** | **76.77** | **39.00** | **69.85** | **65.84** | **62.08** | **35.84** | **57.40** | **59.01** |
| | Wanda | 59.67 | 65.28 | **30.70** | 50.63 | 62.05 | 48.85 | 28.93 | 57.56 | 50.46 |
| | ***SparseGPT w. ALS*** | **60.59** | **65.94** | 29.80 | **50.84** | 61.96 | 47.83 | **29.33** | **57.74** | **50.50** |

**Results in LLaMA-V2 7B/13B at various sparsity level**

Table 15: LLaMA-V2 13B at various sparsity level

| | | winogrande | piqa | openbookqa | hellaswag | boolq | arc_easy | arc_challenge | rte | mean |
|---|---|---|---|---|---|---|---|---|---|---|
| | Magnitude | 49.49 | 53.21 | 26.60 | 29.57 | 38.72 | 32.20 | 24.57 | 52.71 | 38.38 |
| | ***Magnitude w. ALS*** | **53.51** | **62.89** | **30.40** | **40.81** | **60.49** | **41.58** | **25.85** | 52.71 | **46.03** |
| | SparseGPT | 59.27 | **63.71** | **35.20** | **43.81** | 64.07 | **48.44** | 26.88 | 52.71 | 49.26 |
| 70% | ***SparseGPT w. ALS*** | **60.14** | 62.46 | 33.60 | 43.49 | **65.60** | 47.10 | **29.44** | **53.43** | **49.41** |
| | Wanda | 51.38 | 57.18 | **29.80** | 31.65 | 52.69 | **37.21** | 20.73 | **53.07** | 41.71 |
| | ***SparseGPT w. ALS*** | **52.96** | **59.14** | 28.80 | **31.85** | **59.30** | 36.70 | **22.18** | 52.71 | **42.96** |
| | Magnitude | 57.22 | 70.84 | 33.00 | 61.11 | 47.55 | 51.94 | 32.08 | 52.71 | 50.81 |
| | ***Magnitude w. ALS*** | **64.56** | **73.12** | **40.20** | **65.75** | **68.29** | **57.49** | **36.43** | **56.32** | **57.77** |
| | SparseGPT | **68.43** | **75.03** | 40.60 | **66.70** | **76.33** | 62.08 | 38.14 | **58.12** | **60.68** |
| 60% | ***SparseGPT w. ALS*** | 67.96 | 73.56 | **40.80** | 66.50 | 73.91 | **63.76** | **38.57** | 53.43 | 59.81 |
| | Wanda | 67.96 | **76.28** | 40.20 | 65.05 | **78.32** | **65.45** | **40.27** | 56.68 | **61.27** |
| | ***SparseGPT w. ALS*** | **69.69** | 74.37 | **41.20** | **65.43** | 77.83 | 63.30 | 39.33 | 56.68 | 60.98 |
| | Magnitude | 65.27 | 77.20 | 40.60 | 73.01 | 57.68 | 67.17 | 41.89 | **55.96** | 59.85 |
| | ***Magnitude w. ALS*** | **68.35** | **77.48** | **43.60** | **74.42** | **73.15** | **69.91** | **44.63** | 55.60 | **63.39** |
| | SparseGPT | 67.25 | 75.84 | 41.20 | 69.63 | 68.50 | 63.76 | 38.40 | **56.68** | 60.16 |
| 50% | ***SparseGPT w. ALS*** | **71.19** | **78.13** | **44.40** | **74.99** | **81.16** | **69.82** | **43.94** | **63.18** | **65.85** |
| | Wanda | 69.39 | 78.13 | 44.10 | 75.02 | 80.34 | **70.37** | 42.76 | 55.72 | 64.48 |
| | ***SparseGPT w. ALS*** | **72.06** | **78.51** | **45.80** | **75.67** | **81.35** | 70.33 | **46.08** | **62.82** | **66.58** |
| | Magnitude | **70.88** | **78.95** | 45.00 | 77.33 | 74.22 | 74.45 | 47.18 | **59.21** | 65.90 |
| | ***Magnitude w. ALS*** | 70.24 | 78.67 | **45.40** | **77.72** | **76.94** | **75.29** | **48.38** | 54.87 | **65.94** |
| | SparseGPT | **72.38** | **79.82** | 46.00 | **77.77** | 78.75 | 73.23 | **48.46** | 58.48 | **66.86** |
| 40% | ***SparseGPT w. ALS*** | 71.90 | 77.53 | **46.40** | 77.68 | **79.30** | **73.27** | 47.10 | **59.93** | 66.64 |
| | Wanda | **72.22** | **79.54** | 45.60 | 78.55 | 80.92 | **73.70** | 49.06 | 58.48 | 67.26 |
| | ***SparseGPT w. ALS*** | 71.98 | 78.29 | **46.20** | **78.57** | **81.22** | 73.65 | 49.06 | **59.21** | **67.27** |
| | Magnitude | 71.43 | **80.31** | 45.00 | 78.89 | **79.82** | 76.14 | 49.15 | **60.29** | **67.63** |
| | ***Magnitude w. ALS*** | **71.82** | 78.78 | **45.80** | **78.98** | 79.20 | **76.47** | 49.15 | 58.48 | 67.34 |
| | SparseGPT | **73.09** | **79.82** | 45.60 | 78.96 | 79.91 | **76.14** | **50.09** | **62.09** | **68.21** |
| 30% | ***SparseGPT w. ALS*** | 72.77 | 78.78 | **46.40** | **78.96** | **79.97** | 75.88 | 49.23 | 61.73 | 67.97 |
| | Wanda | **71.82** | **79.60** | 46.20 | **79.20** | 80.64 | **76.09** | **50.26** | 60.65 | **68.06** |
| | ***SparseGPT w. ALS*** | 71.59 | 79.11 | **46.40** | 79.07 | **80.86** | 75.55 | 49.91 | **61.01** | 67.94 |

Table 16: LLaMA-V2 7B at various sparsity level

|  |  | winogrande | piqa | openbookqa | hellaswag | boolq | arc_easy | arc_challenge | rte | mean |
|---|---|---|---|---|---|---|---|---|---|---|
| 70% | Magnitude | 49.17 | 51.74 | **28.00** | 26.31 | 37.95 | 27.74 | **27.05** | **53.07** | 37.63 |
|  | *Magnitude w. ALS* | **49.57** | **55.55** | 27.00 | **28.39** | **44.65** | **34.39** | 25.34 | 51.26 | **39.52** |
|  | SparseGPT | **57.46** | 60.28 | 29.00 | 37.73 | **62.66** | 39.98 | 25.34 | 53.07 | 45.69 |
|  | *SparseGPT w. ALS* | 55.64 | **60.83** | **31.40** | **39.03** | 62.54 | **44.32** | 24.74 | **56.32** | **46.85** |
|  | Wanda | **51.70** | 54.52 | 26.60 | **30.13** | 38.93 | 31.57 | 20.90 | 52.71 | 38.38 |
|  | *SparseGPT w. ALS* | 50.20 | **56.37** | **28.60** | 30.01 | **48.10** | **32.37** | **21.76** | 52.71 | **40.02** |
| 60% | Magnitude | 53.20 | 62.46 | 32.60 | 42.93 | 47.71 | 44.23 | 29.44 | 50.90 | 45.43 |
|  | *Magnitude w. ALS* | **58.01** | **66.81** | **35.20** | **51.70** | **63.09** | **51.89** | **31.74** | **58.12** | **52.07** |
|  | SparseGPT | 63.69 | **71.49** | **38.40** | 60.21 | 67.86 | 60.23 | **36.09** | 52.71 | **56.33** |
|  | *SparseGPT w. ALS* | **64.25** | 70.95 | 36.00 | **60.71** | 64.59 | **60.98** | 35.41 | **53.43** | 55.79 |
|  | Wanda | 62.51 | **72.20** | 37.60 | 57.45 | **68.81** | **60.27** | **34.81** | 53.79 | **55.93** |
|  | *SparseGPT w. ALS* | **63.38** | 71.93 | **38.60** | **57.72** | 63.64 | 59.93 | 34.64 | **54.51** | 55.54 |
| 50% | Magnitude | 63.30 | 73.67 | 38.80 | 65.58 | 62.94 | 57.70 | 37.46 | **57.04** | 57.06 |
|  | *Magnitude w. ALS* | **65.19** | **75.46** | **41.40** | **69.11** | **71.38** | **63.47** | **39.51** | 55.24 | **60.09** |
|  | SparseGPT | 63.14 | 71.71 | 35.60 | 63.91 | 69.05 | 58.29 | 34.98 | 54.87 | 56.44 |
|  | *SparseGPT w. ALS* | **67.96** | **76.39** | 40.00 | **70.52** | **70.98** | **67.97** | **41.13** | **55.96** | **61.36** |
|  | Wanda | 67.32 | 76.99 | 41.40 | 68.76 | **75.78** | 69.23 | 41.72 | 53.83 | 61.88 |
|  | *SparseGPT w. ALS* | **67.80** | **77.10** | **44.80** | **70.75** | 75.47 | **69.61** | **42.32** | **54.87** | **62.84** |
| 40% | Magnitude | **69.14** | 76.99 | 42.60 | 73.12 | 70.55 | 69.07 | 43.17 | 55.24 | 62.48 |
|  | *Magnitude w. ALS* | 68.82 | 76.82 | **44.20** | **74.64** | **75.87** | **71.25** | **45.73** | **60.29** | **64.70** |
|  | SparseGPT | **70.64** | **78.46** | 43.40 | 74.28 | **76.91** | 70.79 | 43.94 | 54.51 | **64.12** |
|  | *SparseGPT w. ALS* | 70.17 | 77.20 | **44.00** | **74.81** | 76.61 | **71.00** | **44.03** | 54.51 | 64.04 |
|  | Wanda | **69.30** | **79.33** | 44.20 | 74.35 | 75.72 | 71.89 | **45.82** | 54.87 | **64.43** |
|  | *SparseGPT w. ALS* | 68.75 | 77.53 | **44.60** | **74.65** | **76.12** | **72.18** | 45.22 | **55.23** | 64.29 |
| 30% | Magnitude | 70.64 | 77.48 | **46.40** | 76.10 | 74.25 | 73.49 | 45.90 | 57.40 | 65.21 |
|  | *Magnitude w. ALS* | 70.40 | 78.40 | 44.40 | **76.26** | **77.09** | **73.91** | **47.35** | **59.21** | **65.88** |
|  | SparseGPT | 69.22 | **78.40** | **45.20** | **75.81** | **77.13** | 72.77 | 45.48 | 53.43 | 64.68 |
|  | *SparseGPT w. ALS* | **69.38** | 77.97 | 44.80 | 75.65 | 76.97 | **73.06** | **45.90** | **54.15** | **64.74** |
|  | Wanda | 68.51 | 78.24 | **45.40** | **76.11** | **77.28** | 73.11 | **46.50** | **55.96** | 65.14 |
|  | *SparseGPT w. ALS* | 68.51 | **78.89** | 44.80 | 76.10 | 77.16 | **73.57** | 45.99 | **58.48** | **65.44** |

# D  Visualization of Correlation Matrices

In this section, we present the visualization of correlation matrices obtained by solving the problem under different experimental settings. Additionally, we provide the numerical results of the Sparse Ratio Allocation. The RM heat maps are different with different models.

## D.1  Visualization of RM Matrices

**50% sparsity in LLAMA-V1/V2/V3 family.**

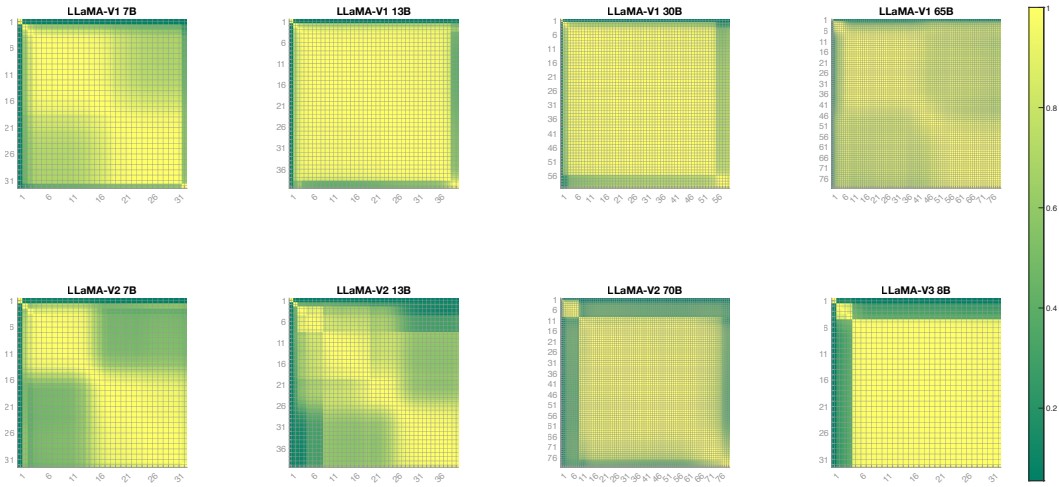

Figure 3: 50% sparsity in LLAMA-V1/V2/V3 family.

**Various sparsity in LLAMA-V2 7B/13B**

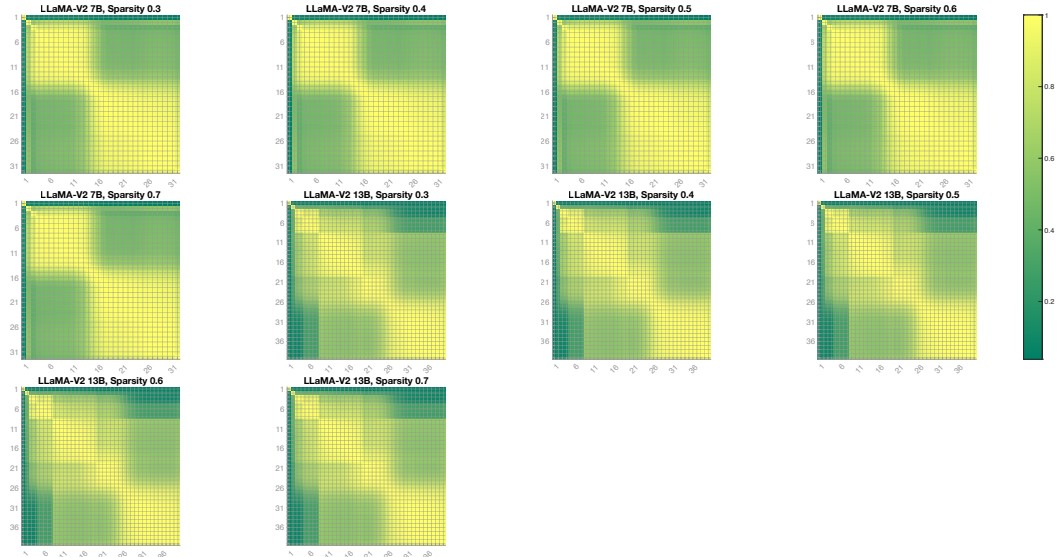

Figure 4: various sparsity in LLAMA-V2 7B/13B family.

## D.2 Ratio allocation

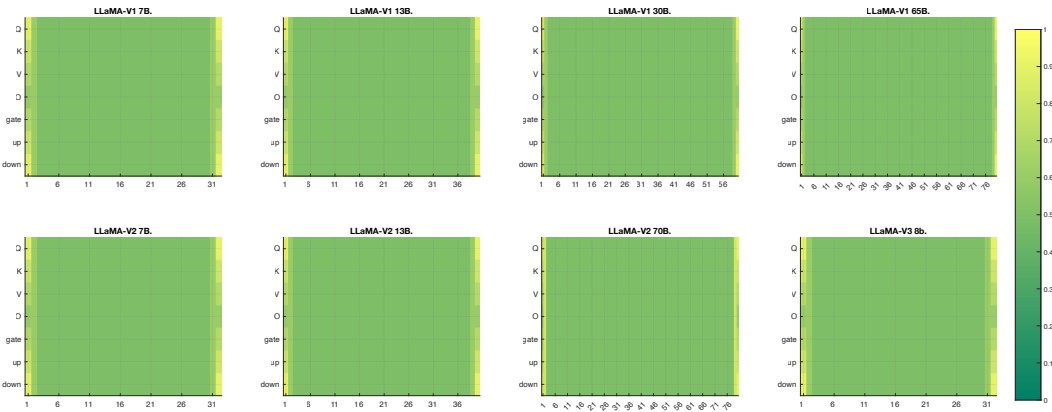

Figure 5: The sparsity ratio allocation in 50% sparsity in LLAMA-V1/V2/V3 family.

## D.3 Ratio allocation

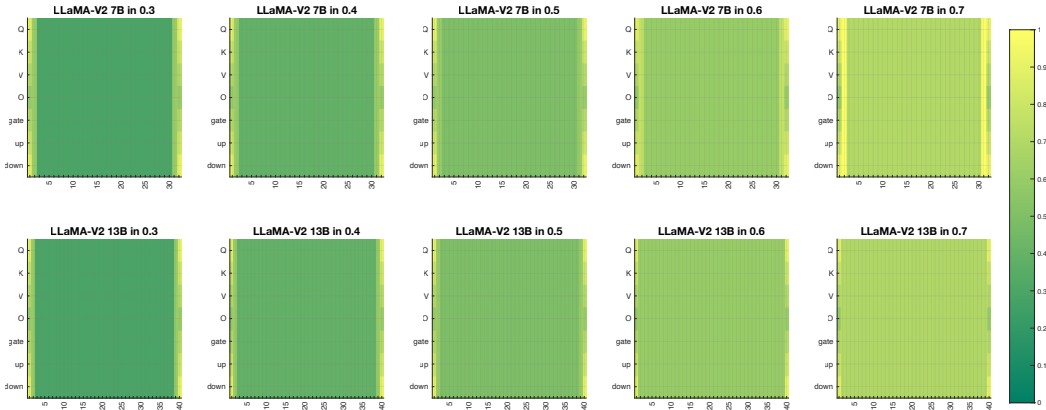

Figure 6: The sparsity ratio allocation in various sparsity in LLAMA-V2 7B/13B family.

# E  Extra Figures and Explanations

## E.1  Experimental Environment and Hyperparameters

**Granularity.** The granularity for linear optimization results is set to 0.5%, meaning the sparsity percentages can only have decimal places of 0.5% or 0.0%. The experiment is based on LLaMA-V2 13B, this study in Fig 7 examines the impact of granularity on perplexity (PPL) across selected values. Initially, PPL remains relatively constant at 10.07 for granularities of 0.1 and 0.5. It then decreases to 9.86 at a granularity of 1 and further to 9.67 at a granularity of 5. However, beyond this point, the smoothed curve indicates a subsequent rise in PPL, suggesting that excessively high granularity may negatively impact model performance. This analysis highlights a critical balance in optimizing granularity to minimize PPL and enhance model accuracy and efficiency.

**Environment.** All pruning experiments are performed on dual NVIDIA A100 GPUs with 80GB memory. However, our ALS method mainly runs on CPU, while the baseline methods Wanda, SparseGPT, and Magnitude require GPU. The CPU used is an AMD EPYC™ 9554 64-core processor.

**Hyperparameters** We set weight and feature normalization, calibration data= 2, feature selection= in, granularity=0.05, boundes= 0.3-0.7 for 3070% sparsity and 0.1-0.3 for 20% sparsity.

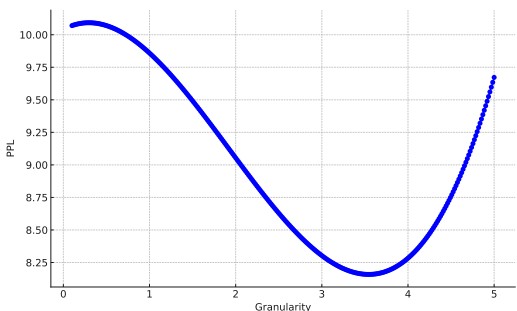

Figure 7: The granularity experiment in LLAMA-V2 7B.

## E.2    Ratio allocation

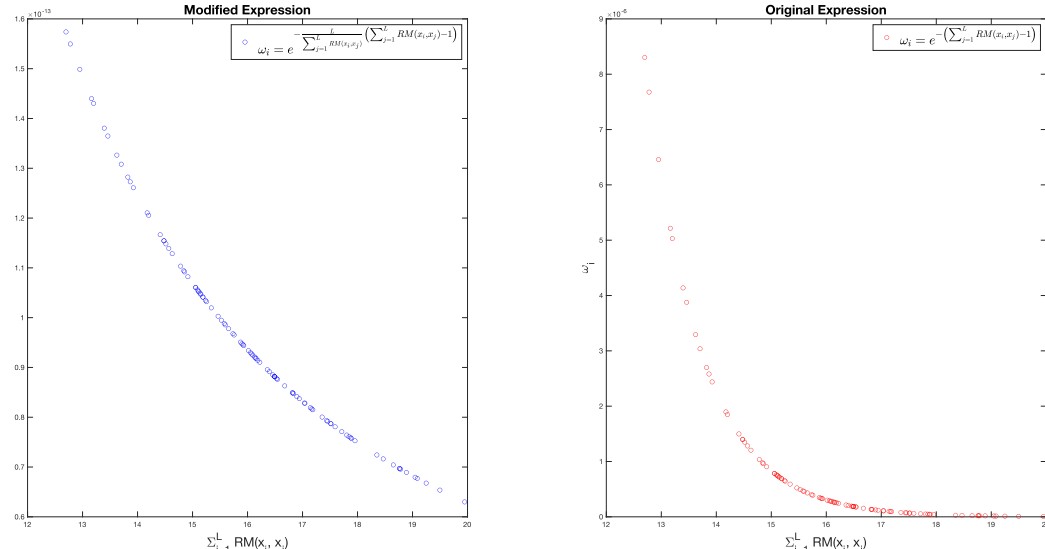

Figure 8: The comparison of decreasing function

## E.3 Error Bar

In this subsection, we present LLaMA-V2 family error bar in 50% sparsity. The standard deviations are runing multiple experiments and get from the EleutherAI LM Harness [47] package.

Table 17: Standard Deviations for Zero-Shot Tasks with 50% Sparsity using LLaMA-V2 Family (Scaled by 100).

| Llama V2 | Method | winogrande | piqa | openbookqa | hellaswag | boolq | arc_easy | arc_challenge | rte |
|---|---|---|---|---|---|---|---|---|---|
| | Magnitude | 1.3804 | 1.0545 | 1.8664 | 0.4969 | 0.8709 | 1.0100 | 1.3787 | 2.9974 |
| | *Magnitude w. ALS* | 1.3692 | 1.0875 | 1.9436 | 0.4987 | 0.8575 | 0.9715 | 1.3952 | 2.9974 |
| | SparseGPT | 1.3308 | 1.0099 | 2.0272 | 0.4989 | 0.7751 | 0.9426 | 1.4125 | 3.0096 |
| 7B | *SparseGPT w. ALS* | 1.3238 | 1.0771 | 1.9966 | 0.4990 | 0.7645 | 0.9346 | 1.4131 | 2.9953 |
| | Wanda | 1.3262 | 1.0047 | 2.0395 | 0.4988 | 0.7765 | 0.9430 | 1.4200 | 2.9882 |
| | *Wanda w. ALS* | 1.3285 | 1.0625 | 2.0144 | 0.4989 | 0.7700 | 0.9419 | 1.4206 | 2.9406 |
| | Magnitude | 1.3540 | 1.0593 | 1.9920 | 0.4956 | 0.8702 | 1.0098 | 1.3831 | 3.0092 |
| | *Magnitude w. ALS* | 1.3107 | 1.0283 | 1.9874 | 0.4988 | 0.8575 | 0.9350 | 1.4206 | 2.9974 |
| | SparseGPT | 1.2759 | 0.9869 | 2.0848 | 0.4974 | 0.7584 | 0.9152 | 1.4426 | 2.9148 |
| 13B | *SparseGPT w. ALS* | 1.2707 | 1.0077 | 1.9966 | 0.4990 | 0.7645 | 0.9346 | 1.4426 | 2.9256 |
| | Wanda | 1.2686 | 0.9679 | 2.0951 | 0.4964 | 0.7474 | 0.9116 | 1.4491 | 2.9308 |
| | *Wanda w. ALS* | 1.2707 | 0.9956 | 2.0144 | 0.4969 | 0.7556 | 0.8997 | 1.4475 | 2.9033 |
| | Magnitude | 1.2224 | 0.9494 | 2.1408 | 0.4864 | 0.7044 | 0.8752 | 1.4611 | 2.9148 |
| | *Magnitude w. ALS* | 1.2224 | 0.9494 | 2.1408 | 0.4864 | 0.7044 | 0.8752 | 1.4611 | 2.9148 |
| | SparseGPT | 1.2173 | 0.9346 | 2.1236 | 0.4878 | 0.6499 | 0.8407 | 1.4581 | 2.9033 |
| 70B | *SparseGPT w. ALS* | 1.2173 | 0.9346 | 2.1236 | 0.4878 | 0.6499 | 0.8407 | 1.4581 | 2.9033 |
| | Wanda | 1.1878 | 0.9298 | 2.1613 | 0.4838 | 0.6216 | 0.8232 | 1.4610 | 2.7073 |
| | *Wanda w. ALS* | 1.1878 | 0.9298 | 2.1613 | 0.4838 | 0.6216 | 0.8232 | 1.4610 | 2.7073 |

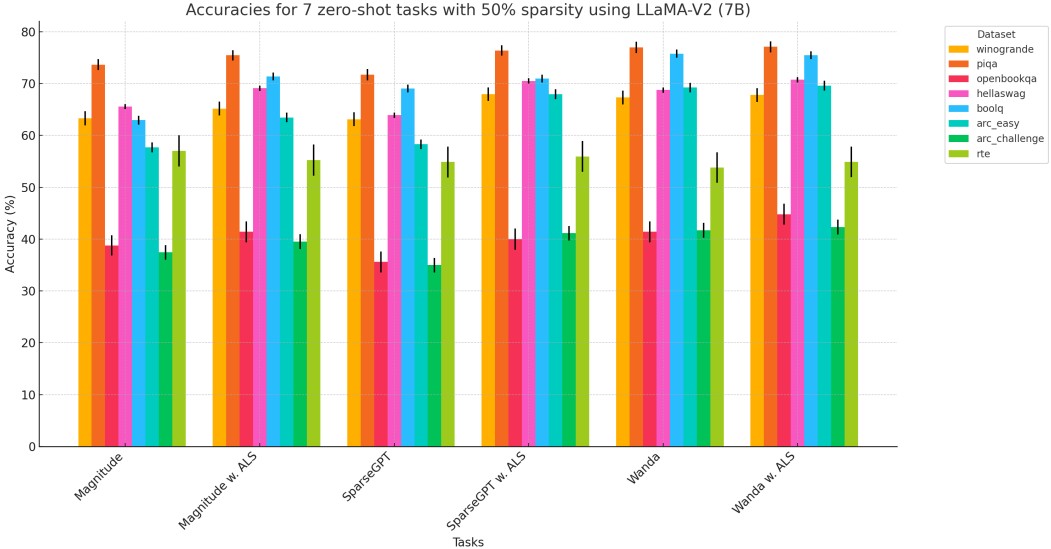

Figure 9: The error bar in 50% sparisity experiment in LLAMA-V2 7B.

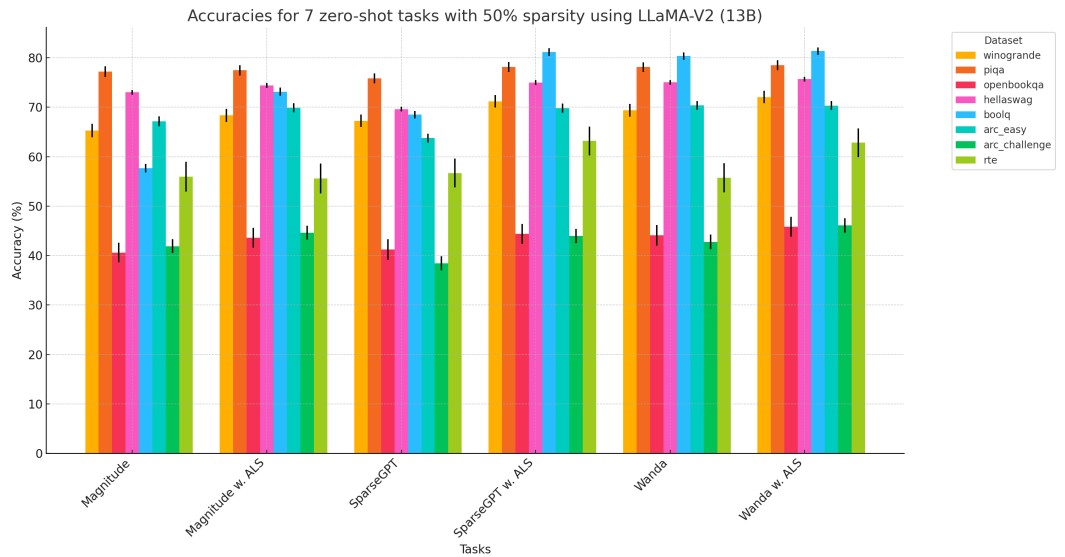

Figure 10: The error bar in 50% sparisity experiment in LLAMA-V2 13B.

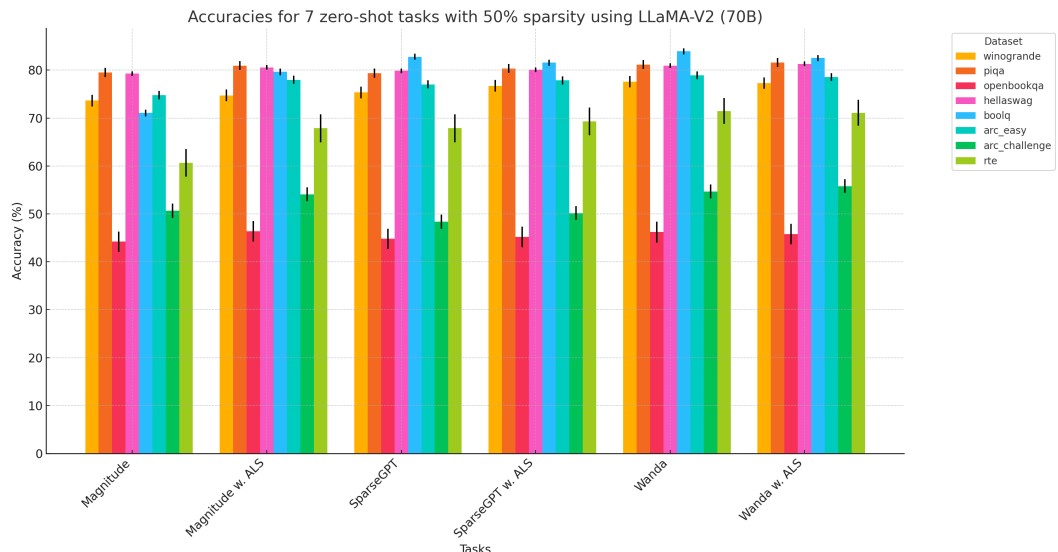

Figure 11: The error bar in 50% sparisity experiment in LLAMA-V2 70B.

