# OpenReview forum: "Adaptive Layer Sparsity for Large Language Models via Activation Correlation Assessment"
_NeurIPS.cc/2024/Conference — NeurIPS 2024 poster_

### Official Review · Reviewer_2MGm · 2024-06-25

**Soundness:** 4
**Presentation:** 4
**Contribution:** 4
**Rating:** 8
**Confidence:** 5

**Summary:**

This paper introduces Adaptive Layer Sparsity (ALS), a novel approach aimed at optimizing large language models (LLMs) through selective pruning. The key contributions of this work include a method that estimates the correlation between intermediate layers using information orthogonality, enabling the precise measurement of each layer's importance. Additionally, the paper formulates the sparsity allocation problem as a linear programming optimization, facilitating efficient global optimization of layer-wise sparsity ratios. Extensive experiments conducted on various LLM families, including LLaMA and OPT, demonstrate consistent performance improvements over existing pruning methods across different model sizes and sparsity levels. The authors also perform analysis experiments to examine the impact of various factors, such as calibration data, sparsity bounds, and feature selection, on the method's performance.

**Strengths:**

1. The paper presents a novel approach for optimizing LLMs. ALS addresses significant challenges in LLM compression, particularly the challenge of manual sparsity setting and suboptimal performance due to uniform pruning ratios.

2. The experimental design of this research is solid and comprehensive. The authors conduct extensive experiments on various LLM families (LLaMA-V1/V2/V3, OPT) with different parameter sizes, providing a robust evaluation of their method's effectiveness.

3. The proposed method, ALS, is simple and effective, and has good computational efficiency.

4. The paper is well-structured and clearly written. The authors provide a detailed explanation of their methodology, including the key steps of estimating the correlation matrix and employing linear optimization for sparse allocation strategy.

**Weaknesses:**

1. There is significant overlap between the introduction and related work sections, which could be consolidated to improve flow and reduce repetition.

2. Some details about ALS, such as normalization and reweighting strategies mentioned in the ablation study section, are not clearly described in the "method" section. These should be elaborated with formulas in the method section.

**Questions:**

See weakness 2. Could you provide some details about the normalization and reweighting strategies mentioned in the ablation study section?

---

> ### Author Rebuttal · Authors · 2024-08-06
>
> Dear Reviewer 2MGm,
>
> Thank you so much for the detailed and constructive comments, and the recognition of the novelty of the proposed method, the writing, and the experimental evaluation on image classification benchmarks. Please see our responses below to your concerns and questions one by one.
>
> ### **Q1**. Overlap between introduction and related work sections:
>
> Our response:
>
> We appreciate the reviewer's keen observation regarding the overlap between our introduction and related work sections.
>
> To address this issue and enhance the overall coherence of our manuscript, we will do  the following reorganization:
>
> **(1).** **Introduction**: We'll streamline this section to focus on:
>
> - Presenting the **core problem** of LLM compression
> - Outlining our **motivation** for developing a new approach
> - Providing a **high-level overview** of our key contributions and approach
>
> **(2).** **Related Work**: Correspondingly, the Related Work section will be expanded and refined to:
>
> - Offer a more **in-depth analysis** of existing methods
> - **Clearly distinguish** our approach from previous work
> - Provide a **critical evaluation** of the strengths and limitations of current techniques
>
> By implementing these changes, we aim to improve the **logical flow** of the paper, eliminate redundancy without sacrificing important content, and offer readers a **clearer roadmap** of the field and our contributions to it.
>
> ### **Q2**. Lack of details on normalization and reweighting strategies
>
> Our response:
>
> Thank you for highlighting the need for more details on our normalization and reweighting strategies.
>
> To address this concern, we'll add a new subsection titled "**Normalization and Reweighting Strategies**" to our methodology section. This addition will include:
>
> **(1).** **Feature Normalization**: We'll detail our approach to standardizing input features across layers, explaining how this helps balance the influence of different feature scales. For example, we'll expand on the process mentioned in Section 3.2, line 185, which introduces feature normalization.
>
> **(2).** **Weight Normalization**: We'll describe our process for normalizing layer weights, which is key to ensuring fair comparisons across the network. We'll elaborate on the statement in Section 3.3, starting from line 220.
>
> **(3).** **Reweighting Mechanism**: We'll introduce our novel reweighting strategy that adjusts the importance of layers based on their position and contribution to the model's output.
>
> **(4).** **Integration with ALS**: Finally, we'll explain how these techniques are seamlessly integrated into the broader ALS framework, demonstrating how they enhance its overall effectiveness.
>
> By including this information, we aim to provide a clearer, more comprehensive picture of our approach.
>
> We appreciate your attention to detail, as it allows us to present a more robust and transparent description of our research.

---

### Official Review · Reviewer_7kYG · 2024-07-05

**Soundness:** 3
**Presentation:** 2
**Contribution:** 2
**Rating:** 5
**Confidence:** 4

**Summary:**

This paper proposes using mutual information to measure layer redundancy in LLMs and employs a linear optimization algorithm based on this measure to derive adaptive sparsity strategies, enabling dynamic sparsity configuration across different layers. The authors conducted experiments on four models, including three versions of LLaMA and OPT, comparing the proposed approach with existing pruning algorithms such as Wanda and SparseGPT. The results demonstrate that the proposed method outperforms these existing algorithms in most scenarios.

**Strengths:**

- The perspective of using mutual information is intuitive and helps identify redundant layers in the model effectively.
- The pruning algorithm is efficient, requiring only a short time to complete the process.
- The experiments on LLaMA-3 are significant, demonstrating the effectiveness of non-uniform sparsity allocation across layers.

**Weaknesses:**

While the combination with ALS effectively improves the baseline performance, there are concerns about the practical value of the method. Specifically, when comparing the pruned models with dense models of similar size, the reported performance is relatively close. An extreme example of this is found in Table 2, where the perplexity (PPL) of the 50% sparse LLaMA-v1 30B model is higher than that of the dense LLaMA-v1 13B model, regardless of the pruning method used. This raises questions about the overall effectiveness of the pruning approach in terms of model efficiency and performance trade-offs.

**Questions:**

- How is the intra-layer pruning calculation performed?
- How is ALS integrated with the baseline method?
- What is the meaning of L-i+1 in equation (5)?
- The analysis in lines 274-276 refers to the 13B model, but the data corresponds to the 7B model in the table, which is confusing. Could you clarify this discrepancy?
- What would be the effect of combining Wanda+ALS with a small amount of LoRA fine-tuning?
- What is the relationship between sparsity and efficiency as studied in this paper? Does it reduce computation time or memory usage?

**Limitations:**

The paper lacks some crucial details in certain areas. For instance, in section 4.3, the process of LoRA fine-tuning is mentioned but not adequately explained. It would be beneficial if the authors could provide more comprehensive information on this aspect.

---

> ### Author Rebuttal · Authors · 2024-08-06
>
> Dear Reviewer 7kYG,
>
> Thank you for your valuable and insightful comments. We have provided detailed responses to your concerns and questions below.
>
> ### **Q1**. Pruned models' performance compared to dense models
>
> Our response:
>
> **(1). Uncommon and Unfair Comparison in Sparsity Research**
>
> - Comparing sparse models to dense models of similar size is **uncommon in sparsity research**. Typically, sparse large models are compared to sparse smaller models, as comparing them to unpruned models is generally considered unfair.
> - Sparse models underperforming similarly-sized dense models is common with SOTA methods like Wanda[41], SparseGPT[13], and Magnitude[26] pruning. This is **not specific to our approach**, but rather a characteristic of current sparsification techniques. This comparison is not the primary focus of our work.
> - Obtaining and post-training sparse large models is **cost-effective** using pre-trained models. For example, 4 GPU hours in A100 80GB of LoRA training achieve **better performance** and **faster inference** than dense models. Training a dense model like Llama2 7B from scratch requires 184,320 GPU hours.
>
>
> **(2). Performance Improvements and Efficiency Gains**
>
> - **Tables 2 and 3** in the paper show that our ALS method **consistently improves performance** across various model sizes and families at 50% sparsity. Notable examples include:
>
>   **(a)**. LLaMA-V3 8B:
>
>   ​	Magnitude pruning: Perplexity reduces by1069 (from 1.1e3 to 30.20)
>
>   ​	SparseGPT: Accuracy increases by 9.82%
>
>   **(b).** LLaMA-V2 13B:
>
>   ​	SparseGPT: Perplexity reduces by 1.88
>
>   ​	Wanda: Accuracy increases by 2.1%
>
> - **Further improvements with LoRA fine-tuning (Table C and D in the gobal rebuttal):**
>
>   **(a).** LLaMA-V1 30B (50% sparse) compared to 13B dense:
>
>   ​	Perplexity: Reduced by **1.21**
>
>   ​	Accuracy: Increased by **2.94%**
>
>   **(b).** LLaMA-V2 13B (50% sparse) compared to 7B dense:
>
>   ​	Perplexity: Reduced by **0.13**
>
>   ​	Accuracy: Increased by **0.53%**
>
> - **Inference acceleration:**
>
>   **(a).** LLaMA-V2 7B at 50% sparsity: 2.355x speedup in inference time
>
>   **(b).** LLaMA-V2 13B (50% sparse) vs 7B dense: Better throughput (0.0118 vs 0.0084 items/sec), Lower   latency (41.24 vs 58.20 ms)
>
> - Significant decrease in memory usage, at 50% sparsity, **47.5% memory reduction** as shown in **Table F** in the global rebuttal.
>
> ### **Q2&Q3**. Intra-layer pruning calculation method and Integration of ALS with baseline methods
>
> Our response:
>
> The integration of **ALS with baselines** involves three main steps, ensuring a **fine-grained** and **adaptive sparsity distribution** at both inter-layer and intra-layer levels:
>
>
> **(1). Calculate adaptive sparsity ratios for blocks**
>
> - Compute the **Redundancy Metric (RM)** matrix for layers (Equation 4)
> - Apply **Linear Optimization** (Section 3.3)
>
> **(2). Calculate adaptive sparsity ratios for sub-blocks**
>
> - Compute RM matrix for intra-layer components
> - Use Linear Optimization for optimal intra-layer sparsity
>
> **(3). Apply sparsity ratios to baseline pruning methods**
>
> - Use the calculated ratios as input for existing pruning techniques
>
> This approach ensures a **hierarchical, adaptive sparsity distribution** throughout the network architecture. The detailed algorithm pipeline is illustrated in **Figure 1** of the paper
>
> *Note: Detailed implementation is in the submitted code, to be released upon paper acceptance*
>
> ### **Q4**. Meaning of L-i+1 in equation (5)
>
> Our response:
>
> To clarify the meaning of **L - i + 1** in equation (5), it's important to note that this term plays a crucial role in calculating each layer's **average independence** with **subsequent layers**:
>
> **Calculation method**: For layer i, we compute its independence with layers i+1 to L, as shown in Appendix B.3, line 595.
>
> **Examples**:
>
> **(1).** Layer 2: average over L-1 layers (divided by L-1). Subsequent layers follow this pattern
>
> **Benefits of this approach**:
>
> This formulation captures the **hierarchical nature** of neural networks and provides **adaptive sparsity allocation**. **Adaptive weighting** prioritizes earlier layers, ensuring **independence** from previous layers and reflecting each layer's importance.
>
> ### **Q5**. Discrepancy in analysis of 13B vs 7B model
>
> Our response:
>
> Thank you for identifying this typo, which should refer to the 7B model. We will fix this typo in the revision.
>
> ### **Q6 & Limitation**. Effect of combining Wanda+ALS with LoRA fine-tuning
>
> Our response:
>
> In addition to the LoRA experiment presented in Table 5 of the paper, we have conducted further experiments. The effect of combining **Wanda+ALS with LoRA fine-tuning** copmaring with **Wanda+ALS**  is significantly positive (**Table C and D** in the global rebuttal).
>
> **(1). For LLaMA-V1** at 50% sparsity: **30B** have PPL reduced by **1.43** and ACC rose by **2.09%**; **13B** have PPL reduced by **1.15** and ACC rise by **1.89%**; **7B** have PPL reduced by **4.82** and ACC rose by 10.17%.
>
> **(2).** **For LLaMA-V2**: **13B** have PPL reduced by **1.56** and ACC rose by **1.23%**; **7B** have PPL reduced by **1.72** and ACC rose by 2.21%.
>
> Notably, LoRA fine-tuning uses 2000 C4 samples (same dataset as Llama pretraining) in a zero-shot setting, unrelated to evaluation tasks.
> ### **Q7**. Relationship between sparsity and efficiency
>
> Our response:
>
> Our study shows **significant improvements** in computation time and memory usage with sparsity:
>
> **(1).** **Computation Time & Memory Usage**, **Table E,F** in the global rebuttal:
>
> - At 50% sparsity: **2.355x speedup, 47.5% memory reduction**
> - At 70% sparsity: **2.729x speedup**, **63.9% memory reduction**
>
> **(2).** **Practical benefits**: Even at moderate sparsity (50%), significant improvements in both speed and memory are achieved
>
> In conclusion, our sparsification approach significantly reduces **computation time and memory usage** without substantial performance loss, while also supporting 2:4 and 4:8 structured pruning (see Appendix C).

---

> > ### Author Response · Authors · 2024-08-11
> > **Look Forward to The Post-Rebuttal Feedback.**
> >
> > Dear Reviewer 7kYG,
> >
> > Thanks for your careful and constructive comments again. We have addressed your concerns in our rebuttal point-by-point. Please let us know if there are any further questions. If our response alleviates your concerns and clarifies the value of our paper, we would be truly grateful if you could reconsider your recommendation. We promise to thoroughly reflect all your comments in the final manuscript. We believe we have faithfully incorporated the feedback from all four reviewers and hope this is reflected positively in your evaluation. Thank you for taking the time to read our response.
> >
> >
> > Best regards,
> >
> > Paper 688 Authors

---

> ### Comment · Reviewer_7kYG · 2024-08-11
>
> Thank you for your detailed response, which has resolved most of my concerns. Therefore, I am raising my score to 5.

---

> > ### Author Response · Authors · 2024-08-11
> > **Thanks for the Positive Feedback and Recognition of Our Work and Rebuttal**
> >
> > Dear Reviewer 7kYG,
> >
> > Thank you so much for the recognition of our responses. We are glad to your positive feedback! Thanks!
> >
> > Following your constructive suggestions, we will make more efforts to improve our paper further.
> >
> > Many thanks for your constructive comments, time and patience.
> >
> > Best regards,
> >
> > Paper 688 Authors

---

### Official Review · Reviewer_ZqVF · 2024-07-11

**Soundness:** 2
**Presentation:** 3
**Contribution:** 3
**Rating:** 6
**Confidence:** 3

**Summary:**

The paper presents an approach called Adaptive Layer Sparsity (ALS) for optimizing large language models (LLMs) by selectively pruning features in intermediate layers. The approach consists of two key steps: estimating the correlation matrix between intermediate layers and employing a linear optimization algorithm to develop an adaptive sparse allocation strategy based on the correlation matrix.

**Strengths:**

1) The paper introduces a novel approach for optimizing LLMs by considering the varying importance of features across different layers.

2) The approach is based on a well-structured methodology, including estimating the correlation matrix and employing a linear optimization algorithm.

3) Compared with BESA which takes 5 hours on an A100-80G GPU, the proposed method takes less time and is more efficient.

**Weaknesses:**

1) From Table 1, it seems that when the sparse ratio is low (e.g., 20%), the performance improvement with ALS is marginal. Does that indicate it is useless for not very sparse models?

2) From Table 2, for stronger models (e.g., 65B and 70B), the improvement is far less than smaller models. Therefore, it is questionable about the applicability of the proposed method to strong models.

3) The authors conducted comprehensive experiments on Llama-series models, but omit Llama-3-80B. Are there any justifications? Since Llama-3-80B is one of the strongest models for now, it would enhance the paper if you can include such results.

**Questions:**

N/A

---

> ### Author Rebuttal · Authors · 2024-08-06
>
> Dear Reviewer ZqVF,
>
> Thank you very much for your detailed and constructive feedback. We will address your concerns and questions as follows.
>
> ### **Q1**. Marginal improvement at low sparsity ratios
>
> Our Response:
>
> At lower sparsity levels (20-30%), improvements from ALS are less pronounced, as even uniform pruning performs well when only a small number of parameters are removed. For example, with **LLaMA-V2-7B** at **20%** sparsity in Table 2 in the paper, perplexity only improves from **8.92 to 8.90** using Wanda w ALS (dense is 8.71), which leave very small room to improve.
>
> However, our method maintains **model stability** across various sparsity levels and model sizes, showing **consistent improvements**. It particularly excels at **high sparsities** crucial for real-world deployment, accelerating inference in resource-constrained environments. For instance, with the LLaMA-V2-13B model (Table 1):
>
> **(1).** At **50% sparsity**: Perplexity reduces by **1.88**
>
> **(2).** At **60% sparsity**: Perplexity reduces by **4.91**
>
> **(3).** At **70% sparsity**: Perplexity dramatically reduces from 1.4e3 to **204.17**
>
> These results demonstrate ALS's effectiveness, particularly in **preventing model collapse**. Specifically:
>
> **(1).** **LLaMA-V2-7B at 60% sparsity**: 83.23 perplexity where Magnitude pruning fails.
>
> **(2).** **LLaMA-V2-13B at 70% sparsity**: 204.17 perplexity where Magnitude pruning fails.
>
> **Llama3 70B** in **Table A** in the global rebuttal, shows that:
>
> **(1).** At **50% sparsity**: Perplexity improves from 8.49 to **7.24** with SparseGPT+ALS
>
> **(2).**  At **60% sparsity**: 13.01 to 12.36
>
> ALS **excels** in scenarios requiring substantial model size reduction, offering a **versatile tool** for efficient LLM deployment across various compression levels. Consequently, these results underscore our approach's **effectiveness**, highlighting its potential for efficient deployment in **resource-limited environments**.
>
> ### **Q2**. Decreased improvement for stronger models
>
> Our response :
>
> We have observed that larger models show less performance improvement with our ALS method. This observation aligns with common patterns in model pruning and compression, where returns tend to diminish as models grow in size. However, our ALS method still shows consistent improvements across different model sizes, especially notable in larger models and at higher sparsity levels:
>
>
>
> **(1). Significance of Relative Gains**
>
> While absolute improvements may appear smaller for larger models, the **relative gains** remain substantial. For Llama3 70B at 50% sparsity, our ALS method improves perplexity from 8.49 to **7.24** with SparseGPT, representing a **14.72% relative improvement**. In the context of large language models, such gains can translate to **meaningful performance enhancements** in real-world applications.
>
> **(2). Inherent Capabilities of Larger Models**
>
> Larger models exhibit **a natural resistance** to pruning. As shown in Table 2 of the paper, the LLaMA-V1 65B model at 50% sparsity only increases in perplexity from 4.93 to 7.37 with Wanda, leaving small room for improvement. Nevertheless, our method further reduces this to 7.15. This observation aligns with the findings of Li et al. [1].
>
> **(3). Effectiveness at Higher Sparsity Levels**
>
> Our method demonstrates particular strength at **higher sparsity levels**, especially for the **Llama3 70B** model, as shown in **Tables A and B** in the global rebuttal. For instance:
>
> - At 70% sparsity, Wanda w. ALS **reduces PPL by 32.98** compared to Wanda alone.
>
> - At 60% sparsity, Magnitude w. ALS **reduces PPL by 341.16** compared to Magnitude alone.
>
> These significant improvements in performance retention at high sparsity levels are crucial for practical deployments where substantial model size reduction is required.
>
> **(4). Alignment with Theoretical Insights**
>
> The behavior we observe aligns with recent theoretical insights. Liu et al. [2] demonstrated that larger models can maintain performance even under **random pruning**, supporting the idea of their inherent pruning resistance.
>
> In essence, the ALS method remains **effective across all model sizes**, offering crucial stability and performance benefits, especially at higher sparsity levels.
>
>
> ### **Q3**. Omission of Llama-3-80B in experiments
>
> Our response
>
> We have conducted comprehensive experiments on the Llama3 70B model. Our findings, detailed in **Tables A, B and pdf** in the global rebuttal, offer valuable insights into our method's performance on SOTA large language models.
>
> **Overall Trends:**
>
> **(1).** Our ALS method consistently outperforms other pruning methods (Magnitude, SparseGPT, and Wanda) across all sparsity levels (50%, 60%, and 70%).
>
> **(2).** The improvements are particularly pronounced at higher sparsity levels.
>
> **Significant Improvements:**
>
> **(1).** **Extreme Perplexity Reduction at 60% Sparsity:**
>
> - Magnitude pruning: PPL reduces by **341.16** (a remarkable 93.5% reduction)
>
> **(2).** **Consistent Improvements for Top Performers:**
>
> - At 50% sparsity, SparseGPT's PPL reduces by **1.25** (14.7% reduction)
> - At 70% sparsity, Wanda's PPL reduces by **32.98** (21.2% reduction)
>
> **(3).** **Robustness at 70% Sparsity:**
>
> - Magnitude pruning: PPL reduced from 95886.56 to **35844.32** (62.6% improvement). ALS significantly improves performance in challenging scenarios.
>
> In conclusion, our ALS method demonstrates **consistent improvements** across various sparsity levels and particularly **excels** at **higher sparsities**, maintaining reasonable performance even at 70% sparsity.
>
> Reference:
>
> [1] Li, Z., et al. (2020). Train big, then compress: Rethinking model size for efficient training and inference of transformers. ICML 2020.
>
> [2] Liu, S., et al. (2022). The unreasonable effectiveness of random pruning: Return of the most naive baseline for sparse training. arXiv:2202.02643.

---

### Official Review · Reviewer_8iME · 2024-07-12

**Soundness:** 3
**Presentation:** 2
**Contribution:** 2
**Rating:** 6
**Confidence:** 2

**Summary:**

The growing size of LLMs make deployment increasingly challenging. Traditional pruning methods underperform due to uniform strategies that ignore varying feature importance across layers. The authors introduces a new Adaptive Layer Sparsity (ALS) approach that estimates inter-layer correlations using information orthogonality, accurately evaluating each layer's significance. It then employs linear optimization to selectively prune features in intermediate layers, achieving precise model optimization. Experiments demonstrate superior efficiency of ALS over existing sparse models like Wanda and SparseGPT, especially at high sparsity levels.

**Strengths:**

1. The paper covers a wide range of related works and discusses the pros and cons of each work, providing a comprehensive background of this work.
2. Formulating the adaptive pruning with the use of a correlation matrix to measure the importance of each layer and a linear optimization algorithm makes the algorithm itself generalizable and theoretical sound.

**Weaknesses:**

1. The presentation of this paper should be improved. Long paragraphs with various discussion points are mixed together, which is a major weakness of this paper.
2. Evaluation could be improved with numbers on the inference efficiency comparison.

**Questions:**

1. What is the computation efficiency of ALS-pruned model compared with existing baslines?
2. What is the overhead of solving the linear programming problem?
3. It seems the performance improvement of ALS decreases when the model becomes large. Is there any reasons for that?

**Limitations:**

The authors have addressed the limitations and there is no potential negative societal impact of this work.l

---

> ### Author Rebuttal · Authors · 2024-08-06
>
> Dear Reviewer 8iME,
>
> Thank you so much for the detailed and constructive comments. Please see our responses below to your concerns and questions one by one.
>
> ### **Q1**. Improving the paper's presentation.
>
> Our Response:
>
> We appreciate the reviewer's suggestions about writing. Based on these recommendations, we commit to carefully revising each paragraph. As part of this revision, the paper's clarity will improve through several changes as follows. Long sections will be divided into **concise paragraphs**, each beginning with an **informative topic sentence**. To guide readers, we'll introduce strategic subheadings such as **normalization techniques** and **reweighting strategies**. Furthermore, we'll create **smooth transitions** between paragraphs and sections, ensuring a logical flow of ideas throughout the paper.
>
> ### **Q2**.  Evaluation of inference efficiency.
>
> Our Response:
>
> We've conducted experiments to address concerns. **Table E** in the global rebuttal showcases results for the **LLaMA-V2 7B model** across various sparsity levels:
>
> | Sparsity| Latency (ms) | Speedup|
> | ---------- | ------------ | ------- |
> | 0% (Dense) | 58.20        | 1x      |
> | 50%        | 24.71        | 2.355x  |
> | 70%        | 21.32        | 2.729x  |
> | 80%        | 20.49        | 2.840x  |
>
> Results show that **50% sparsity** achieves a **2.355x speedup**, reducing latency from 58.1968 ms to 24.7099 ms.
>
> These results highlight our method's ability to maintain model performance while delivering **substantial computational gains**.
>
> Notably, at **50% sparsity**, we observe a **significant reduction in memory usage**, as the **dense model uses 27.6 GB**, while the **50% sparse model** uses only **13.8 GB**, achieving a 50% reduction. For full details, please refer to **Table F** in the global rebuttal.
>
> ### **Q3&Q4**.  Computation efficiency compared to existing baselines and overhead of solving the linear programming problem
>
> Our Response:
>
> Our method's computational efficiency is demonstrated through experiments on the **LLaMA V2 7B** model. The total cost of our approach includes two main components: **Redundancy Metric (RM)** calculation (90 seconds) and **Linear Programming (LP)** solution (160 milliseconds). For Wanda/SparseGPT/Magnitude w ALS, these times are added to their respective pruning times.
>
> |               | RM (s) | LP (ms) | Magnitude/Sparsegpt/Wanda (s) | Total (min)|
> | --------------- | :---------------------: | :-----------------------: | :-----------------------------: | :-----------: |
> | Magnitude w ALS | 88.59                 | 169                     | 1.62                          | 1.51        |
> | Sparsegpt w ALS | 91.32                 | 158                     | 1058                          | 19.16       |
> | Wanda w ALS     | 89.47                 | 160                     | 199                           | 4.81        |
>
> In comparison, **BESA** [50] requires **4.5 hours** for sparsity allocation and pruning, while our approach is **significantly faster**, completing the process in minutes rather than hours.
>
> For acceleration, we optimize calculations for large models (81 blocks). We use an 80 x 80 RM matrix plus 7 additional 7 x 7 matrices, instead of a large 7 (components within a block) x 80 (blocks) x 7 x 80 matrix.
>
> ### **Q5**. Performance improvement decrease for larger models
>
> Our response :
>
> The apparent decrease in performance improvement for larger models is an interesting trend we’ve observed with our ALS method. This aligns with broader patterns in model pruning and compression, where diminishing returns are common as models grow larger. However, despite this general trend, our ALS method demonstrates consistent performance improvements across various model sizes, with particularly notable effects on larger models and at higher sparsity levels:
>
> **(1). Significance of Relative Gains**
>
> While absolute improvements may appear smaller for larger models, the **relative gains** remain substantial. For Llama3 70B at 50% sparsity, our ALS method improves perplexity from 8.49 to **7.24** with SparseGPT, representing a **14.72% relative improvement**. In the context of large language models, such gains can translate to **meaningful performance enhancements** in real-world applications.
>
> **(2). Inherent Capabilities of Larger Models**
>
> Larger models exhibit **a natural resistance** to pruning. As shown in Table 2 of the paper, the LLaMA-V1 65B model at 50% sparsity only increases in perplexity from 4.93 to 7.37 with Wanda, leaving small room for improvement. Nevertheless, our method further reduces this to 7.15. This observation aligns with the findings of Li et al. [1].
>
> **(3). Effectiveness at Higher Sparsity Levels**
>
> Our method demonstrates particular strength at **higher sparsity levels**, especially for the **Llama3 70B** model, as shown in **Tables A and B** in the global rebuttal. For instance:
>
> - At 70% sparsity, Wanda w. ALS **reduces PPL by 32.98** compared to Wanda alone.
>
> - At 60% sparsity, Magnitude w. ALS **reduces PPL by 341.16** compared to Magnitude alone.
>
> These significant improvements in performance retention at high sparsity levels are crucial for practical deployments where substantial model size reduction is required.
>
> **(4). Alignment with Theoretical Insights**
>
> The behavior we observe aligns with recent theoretical insights. Liu et al. [2] demonstrated that larger models can maintain performance even under **random pruning**, supporting the idea of their inherent pruning resistance.
>
> In essence, the ALS method remains **effective across all model sizes**, offering crucial stability and performance benefits, especially at higher sparsity levels.
>
> Reference:
>
> [1] Li, Z., et al. (2020). Train big, then compress: Rethinking model size for efficient training and inference of transformers. ICML 2020.
>
> [2] Liu, S., et al. (2022). The unreasonable effectiveness of random pruning: Return of the most naive baseline for sparse training. arXiv:2202.02643.

---

> > ### Comment · Reviewer_8iME · 2024-08-14
> >
> > Thank you for the response and including the numbers on the cost of solving the LP problem. It appears to be a significant improvement over the baseline.

---

### Author Rebuttal · Authors · 2024-08-06

Dear Reviewers, Area Chairs, and Program Chairs,

We sincerely thank all four reviewers for their feedback and constructive comments. In the initial review, 3 Accept ratings are given. Reviewers have acknowledged the **novelty**, **theoretical soundness**, **impact**, **efficiency**, **comprehensive evaluation**, **performance**, and **clear presentation** of our work.

**[Novelty]:**

-  Reviewer ZqVF: Novel LLM optimization considering layers
-  Reviewer 2MGm: Novel LLM optimization approach

**[Theoretical Soundness]:**

-  Reviewer 8iME: Adaptive pruning generalizable, theoretically sound
-  Reviewer ZqVF: Well-structured methodology

**[Impactful]:**

-  Reviewer 2MGm: Addresses LLM compression challenges effectively
-  Reviewer 7kYG: Intuitive redundant layer identification method
-  Reviewer 8iME: ALS evaluates layer significance accurately

**[Efficiency]:**

-  Reviewer ZqVF: More efficient than BESA
-  Reviewer 7kYG: Efficient, quick pruning algorithm process
-  Reviewer 2MGm: Simple, effective, efficient method
-  Reviewer 8iME: Superior efficiency at high sparsity

**[Performance]:**

-  Reviewer 7kYG: Effective non-uniform sparsity allocation demonstrated
-  Reviewer ZqVF: Outperforms existing algorithms in scenarios
-  Reviewer 2MGm: Outperforms existing algorithms in scenarios

**[Comprehensive Evaluation]:**

-  Reviewer 2MGm: Solid, comprehensive experimental design conducted
-  Reviewer 8iME: Superior efficiency over existing models
-  Reviewer ZqVF: Comprehensive experiments on Llama-series models

**[Clear Presentation]:**

-  Reviewer 8iME: Comprehensive background of related works
-  Reviewer 2MGm: Well-structured, clear methodology explanation provided

We've conducted additional experiments to address comments. We'll revise our manuscript to strengthen our work and address concerns.

Best wishes,

Authors

---
Additional Experiments (details  in the pdf file):

### Llama3 70B ALS experiments series

#### Table A: Wikitext

| Sparsity               | 50%      | 60%       | 70%       |
| ---------------------- | -------- | --------- | --------- |
| Dense                  | 2.92     | 2.92      | 2.92      |
| Magnitude              | 19.29    | 364.92    | 95886.56  |
| ***Magnitude w. ALS*** | 13.21    | 23.76     | 35844.32  |
| SparseGPT              | 8.49     | 13.01     | 50.937    |
| ***SparseGPT w. ALS*** | 7.24     | 12.36     | **49.14** |
| Wanda                  | 7.01     | 10.41     | 155.603   |
| ***Wanda w. ALS***     | **6.82** | **10.05** | 122.62    |

#### Table B: Averaged accuracies(%)

| Sparsity               | 50%       | 60%       | 70%       |
| ---------------------- | --------- | --------- | --------- |
| Dense                  | 75.43     | 75.43     | 75.43     |
| Magnitude              | 51.28     | 44.50     | 39.62     |
| ***Magnitude w. ALS*** | 53.64     | 45.97     | 39.89     |
| SparseGPT              | 70.26     | 58.66     | **43.83** |
| ***SparseGPT w. ALS*** | 71.12     | 58.61     | 40.34     |
| Wanda                  | 72.25     | 66.03     | 40.56     |
| ***Wanda w. ALS***     | **73.12** | **66.34** | 42.4      |

### Llama-V1,V2 ALS w Lora experiments series

#### Table C: Wikitext

| Models                    | V1-7B    | V1-13B   | V1-30B   | V2-7B    | V2-13B  |
| ------------------------- | -------- | -------- | -------- | -------- | ------- |
| Dense                     | 9.38     | 8.2      | 6.09     | 8.71     | 7.68    |
| Wanda                     | 13.3     | 10.9     | 8.74     | 12.31    | 11.21   |
| Wanda w. ALS              | 12.47    | 10.4     | 8.42     | 11.61    | 9.86    |
| ***Wanda w. ALS & Lora*** | **7.65** | **9.25** | **6.99** | **9.89** | **8.3** |

#### Table D: Averaged accuracies(%)

| Models                    | V1 7B     | V1 13B    | V1 30B    | V2 7B | V2 13B    |
| ------------------------- | --------- | --------- | --------- | ----- | --------- |
| Dense                     | 66.18     | 68.5      | 71.36     | 66.21 | 68.76     |
| Wanda                     | 58.87     | 64.74     | 68.54     | 61.88 | 64.48     |
| Wanda w. ALS              | 61.47     | 64.82     | 69.35     | 62.84 | 66.58     |
| ***Wanda w. ALS & Lora*** | **71.64** | **66.71** | **71.44** | **65.05** | **67.81** |

### Latency Experiment

#### Table E: Llama V2 7B

| Sparsity               | Dense   | 30%    | 40%     | 50%     | 60%     | 70%     | 80%     |
| ---------------------- | ------- | ------ | ------- | ------- | ------- | ------- | ------- |
| Latency (ms)           | 58.1968 | 58.136 | 33.0388 | 24.7099 | 23.4007 | 21.3224 | 20.4892 |
| Throughput (items/sec) | 0.0084  | 0.0084 | 0.0148  | 0.0198  | 0.0209  | 0.0229  | 0.0238  |
| Speedup                | 1x      | 1.001x | 1.761x  | 2.355x  | 2.487x  | 2.729x  | 2.840x  |

### Memory Usage with unstructured Sparsity for Llama V2-7B

#### Table F: Memory usage

| Configuration | Memory Usage (GB) |
| ------------- | ----------------- |
| FP32 Dense    | 26.3              |
| FP32 30%      | 19.1              |
| FP32 40%      | 16.2              |
| FP32 50%      | 13.8              |
| FP32 60%      | 11.9              |
| FP32 70%      | 9.5               |

---

### Decision · Program_Chairs · 2024-09-25

**Decision:**

Accept (poster)

**Comment:**

The paper introduces Adaptive Layer Sparsity, a method to optimize LLMs by selectively pruning less important features in intermediate layers based on a correlation matrix, thereby reducing model size without compromising performance. Through extensive experiments, ALS demonstrates good performance compared to existing methods.

The paper is well-written and easy to follow. The reviewers had some questions and concerns about the experiments, technical details, and the motivation behind the paper. During the rebuttal, the authors addressed these issues, and the reviewers were satisfied with the responses and additional experiments. Therefore, I recommend accepting the paper.